# Persuasive Privacy

**Joshua J Bon**[1]   **James Bailie**[2]   **Judith Rousseau**[3]   **Christian P Robert**[3 4]

## Abstract

We propose a novel framework for measuring privacy from a Bayesian game-theoretic perspective. This framework enables the creation of new, purpose-driven privacy definitions that are rigorously justified, while also allowing for the assessment of existing privacy guarantees through game theory. We show that pure and probabilistic differential privacy are special cases of our framework, and provide new interpretations of the post-processing inequality in this setting. Further, we demonstrate that privacy guarantees can be established for deterministic algorithms, which are overlooked by current privacy standards.

## 1. Introduction

The scientific and economic value of data grows alongside technological advances. New hardware and software developments enable, but often require, larger and more complex datasets to function effectively. As the importance of input data to these systems becomes increasingly recognized, so too does the loss of privacy for data providers. In this context, data privacy emerges as a critical issue for fields such as statistics and machine learning, as well as for scientific and industrial endeavours that rely on sensitive data.

In the last two decades, differential privacy (DP, Dwork et al., 2006b;a; Dwork, 2006) and its variants (see Desfontaines & Pejó, 2020) have become the de facto standard for data privacy. However, DP continues to face conceptual and practical challenges, including difficulties in interpreting and communicating its parameters (Cummings & Sarathy, 2023); gaps between its implementation and legal or social notions of privacy (Seeman & Susser, 2024);

and the large, potentially vacuous, privacy loss budgets often seen in its real-world deployments (Dwork et al., 2019; Schneider et al., 2025). With these concerns in mind, we develop a framework to (i) generate privacy definitions that are fit for purpose yet rigorously justified; (ii) assess existing privacy guarantees using a Bayesian game-theoretic approach; and (iii) evaluate the privacy of deterministic algorithms. This latter capability is incompatible with DP and most of its variants (see Appendix D.3), and is particularly important to the problem of assessing the privacy leakage of invariant statistics—deterministic summaries of the sensitive data which are integral to many types of statistical dissemination, including the US Decennial Census (Abowd et al., 2022; Bailie et al., 2026c).

Our new framework for privacy is developed rigorously and systematically using game theory. We make explicit assumptions to establish our framework and discuss their merits and necessity. Our approach allows for a semantics-first understanding of data privacy, where each assumption can be tested against real-world considerations, and provides privacy definitions which are easy to interpret, communicate and tailor because the framework is constructed from an agent-based game. In contrast, semantic interpretations of DP have been constructed post-hoc (Kasiviswanathan & Smith, 2014; Wasserman & Zhou, 2010) and can be difficult to understand (Nanayakkara et al., 2023; Cummings et al., 2021). We address this limitation whilst facilitating future work to relax or modify our assumptions when they are not well suited to a particular use case.

We show that privacy definitions generated by our framework satisfy a composition property and a weaker version of the standard post-processing inequality. We discuss an agent-based interpretation of the standard post-processing inequality, and show how it is trivial to ensure that it holds in practice. We show how our framework encompasses pure $\varepsilon$-DP and probabilistic $(\varepsilon, \delta)$-DP, whilst a minor change establishes a formal connection to Rényi DP (Mironov, 2017) and $f$-divergence privacy (Barber & Duchi, 2014; Barthe & Olmedo, 2013). Further comparisons to existing privacy definitions, including quantitative information flow (Alvim et al., 2020) and pufferfish privacy (Kifer & Machanava-jjhala, 2014), are discussed in Appendix D.

[1]School of Mathematical Sciences, Adelaide University, Australia [2]Department of Computer Science and Engineering, Chalmers University of Technology and University of Gothenburg, Sweden [3]CEREMADE, Université Paris–Dauphine PSL, France [4]Department of Statistics, University of Warwick, UK. Correspondence to: Joshua J Bon <joshuajbon@gmail.com>.

*Proceedings of the $43^{rd}$ International Conference on Machine Learning*, Seoul, South Korea. PMLR 306, 2026. Copyright 2026 by the author(s).

## 1.1. Related Work

Statistical data privacy (Slavković & Seeman, 2023) has an extensive literature which stretches back to at least the 1970s (Dalenius, 1977) and includes both modern, formal theories (e.g., DP) as well as traditional statistical disclosure control (SDC, Hundepool et al., 2012; Willenborg & de Waal, 2001). Although the problem of data privacy is typically not expressed in terms of game theory, this field nevertheless shares with our work the concepts of Sender (the agency releasing statistics) and Receiver (the adversary seeking to exploit the published data). The concern in statistical data privacy is to limit Receiver's ability to learn about individual data points from the published statistics (Duncan & Lambert, 1986; Dwork & Naor, 2010); we generalise this with a "privacy function" which represents Sender's loss (in an abstract sense) due to Receiver's decision. This is similar to Bun et al. (2025) which frames privacy in terms of Receiver's ability to inflict harm through the use of the published data.

As with our work, Bayesian formulations of Receiver are common in both DP (see e.g., Dwork et al., 2006b; Kifer & Machanavajjhala, 2014; 2011; Kasiviswanathan & Smith, 2014; Kifer et al., 2022) and traditional SDC (see e.g., Fienberg et al., 1997; Dobra et al., 2003). However, we are, to the best of our knowledge, the first to consider Sender as a player in the game, similar to the set-up of Bayesian Persuasion (Kamenica & Gentzkow, 2011).

Several lines of work connect DP and game theory, ranging from game-theoretic DP parameter settings (Kohli & Laskowski, 2018; Hsu et al., 2014) to the use of DP for mechanism design (McSherry & Talwar, 2007; Nissim et al., 2012; Pai & Roth, 2013) as well as the pricing personal data with DP (Ghosh & Roth, 2015; Dandekar et al., 2014; Roth & Schoenebeck, 2012; Fleischer & Lyu, 2012; Ligett & Roth, 2012; Li et al., 2017). There is also literature extending existing games, such as Bayesian Persuasion, to include DP constraints (Pan et al., 2025). In contrast, we provide a fully game-theoretic foundation and justification of DP, amongst other privacy definitions.

## 1.2. Notation

We use the terms *privacy definition* and *guarantee* interchangeably, and refer to the set of all data release mechanisms satisfying a certain privacy definition as a *privacy class*. A given privacy definition will generate a privacy class. For example, the set of all mechanisms satisfying $\varepsilon$-DP is a privacy class. The sensitive dataset is denoted by $x$. We consider $\mathsf{X}$ to be some universe of possible datasets, such that $x \in \mathsf{X}$. We use $z$ as a dummy variable in place of $x$ if distinction or generality is required.

Probability distributions $P, Q : \mathcal{X} \to [0, 1]$ expressing un-

certainty about the value of $x \in \mathsf{X}$ are defined on a measurable space $(\mathsf{X}, \mathcal{X})$, and a family of such distributions is denoted by $\mathcal{P}$. For example, $P = \mathcal{N}(\mu, \Sigma)$ may express the uncertainty held by a player about the data $x \in \mathbb{R}^n$. Expectations of $f : \mathsf{X} \to \mathbb{R}$ under $P$ are written as $\mathbb{E}_{X \sim P}[f(X)]$.

We denote Markov kernels by $M : (\mathsf{Z}, \mathcal{T}) \to [0, 1]$ for input and output measurable spaces[1] $(\mathsf{Z}, \mathcal{Z})$ and $(\mathsf{T}, \mathcal{T})$, respectively. That is, $M(z, \cdot)$ is a probability distribution on $(\mathsf{T}, \mathcal{T})$ for a given $z \in \mathsf{Z}$. We use Markov kernels to represent data release mechanisms. For example, releasing $\bar{x}$ with additive Gaussian noise can be written as $M(x, \cdot) = \mathcal{N}(\bar{x}, \sigma^2)$. Deterministic mechanisms can be expressed as $M(x, \cdot) = \delta_{f(x)}$ for a function $f$, where $\delta_t$ is a degenerate probability distribution with point mass at $t$. The identity kernel will be denoted by $\mathrm{Id}$. The probability of an event $E \in \mathcal{T}$ under a mechanism $M(x, \cdot)$ is denoted by $\mathbb{P}_x(E)$, conditional on $x \in \mathsf{X}$. The composition (resp. tensor product) of two Markov kernels, say $M$ and $K$, is denoted by $MK$ (resp. $M \otimes K$). For $M : (\mathsf{Z}, \mathcal{T}_1) \to [0, 1]$ and $K : (\mathsf{Z} \times \mathsf{T}_1, \mathcal{T}_2) \to [0, 1]$, the composition and tensor product are defined as

$$MK(z, \cdot) = \int M(z, \mathrm{d}t_1) K((z, t_1), \cdot) \text{ and}$$

$$(M \otimes K)(z, \mathrm{d}(t_1, t_2)) = M(z, \mathrm{d}t_1) K((z, t_1), \mathrm{d}t_2),$$

respectively, for fixed $z \in \mathsf{Z}$. We say that a Markov kernel $M$, defined by $M((x, y), \cdot)$ for all $(x, y) \in \mathsf{X} \times \mathsf{Y}$, is independent of $x$ if $M((x_1, y), \cdot) = M((x_2, y), \cdot)$ for all $x_1, x_2 \in \mathsf{X}$.

We denote regular conditional probability distributions (see e.g., Durrett, 2019) using conditioning notation. For example, if $P$ is a prior distribution for $x$ then $P(\cdot \mid T)$ denotes the posterior distribution, after observing $T \sim M(x, \cdot)$.

We make use of proper scoring rules (see Gneiting & Raftery, 2007, for an overview). Let $S : (\mathcal{P}, \mathsf{X}) \to \mathbb{R} \cup \{-\infty, \infty\}$ be a (negatively-orientated) scoring rule. Scoring rules measure the success of predictive distribution $P$ to estimate the truth $x$ by $S(P, x)$, where $S(P, x) < S(Q, x)$ if $P$ outperforms $Q$ (according to $S$). Later, we show how *proper* scoring rules arise naturally in our framework. Let $S(P, Q) = \mathbb{E}_{X \sim Q}[S(P, X)]$ be the *expected score* under $Q$.

**Definition 1.** A scoring rule is proper if $S(Q, Q) \leq S(P, Q)$ for all $P, Q \in \mathcal{P}$. A strictly proper scoring rule is proper and $S(Q, Q) = S(P, Q)$ if and only if $P = Q$.

## 2. Privacy as Persuasion

We propose a two-player Stackelberg game, involving Sender and Receiver, to construct a new class of privacy

---

[1]When multiple kernels are required we will also denote measurable spaces by $(\mathsf{T}_k, \mathcal{T}_k)$ for positive integers $k$.

definitions. Sender is the custodian of the sensitive data $x \in \mathsf{X}$ and designs a mechanism $M$ to release useful information derived from the data. Sender shares the (potentially stochastic) output of the mechanism with Receiver, who then takes an action (or makes a decision) that will affect the privacy of Sender.

Our game-theoretic foundation of the privacy framework is closely related to Bayesian Persuasion (Kamenica & Gentzkow, 2011). Ours contrasts this work in three main ways. Firstly, we model an information asymmetry between Sender and Receiver. That is, Sender knows the true value of the sensitive data, whilst Receiver has uncertainty about the data expressed as a prior distribution. Secondly, the utility functions of Sender and Receiver are related. Lastly, we assume that Sender assesses decisions (i.e., the mechanism to choose) in a robust manner by assessing worst-case, rather than expected, outcomes for privacy. For further discussion of Bayesian Persuasion, also known as information design, see Kamenica (2019).

## 2.1. Preliminaries

Sender assesses the level of privacy based on the context and goals related to sharing information derived from the data. They consider adversarial decisions (actions) made by Receiver in a decision space $\mathcal{D}$ which determines privacy relative to the value of the data.

### 2.1.1. PRIVACY FUNCTIONS

We define Sender's privacy relative to decision $d \in \mathcal{D}$ and data value $x \in \mathsf{X}$ as a privacy function.

**Definition 2** (Privacy function)**.** A privacy function $\rho : (\mathcal{D}, \mathsf{X}) \to \mathbb{R}$ represents the preferences of Sender toward Receiver's possible decisions $d_i \in \mathcal{D}$. For $x \in \mathsf{X}$, if $d_1$ is preferred to $d_2$, then $\rho(d_1, x) > \rho(d_2, x)$.

Privacy is positively-oriented (for Sender) and hence higher values of $\rho(d, x)$ indicate higher privacy under value $x$, for an adversarial decision $d$. Sender may assess privacy with one or more privacy functions, an extension we consider in Section 3. We focus on the case of one privacy function for ease of exposition in this section.

**Example 1.** Let $\mathcal{D} = \{[a, b] : a, b \in \mathbb{R}, a \leq b\}$ and $\mathsf{X} = \mathbb{R}$. Given $s > 0$, the interval privacy function is defined as

$$\rho(d, x) = \begin{cases} 0 & \text{if } x \in d \text{ and } |d| \leq s, \\ 1 & \text{otherwise.} \end{cases}$$

In this example, privacy is a binary outcome. Privacy is achieved by Sender if the true data $x$ is not contained in Receiver's decision, an interval, or if the interval is sufficiently large.

**Example 2.** Let $\mathcal{D}$ be the set of probability density functions on $\mathsf{X} = \mathbb{R}$ with respect to some given measure $\mu$. For $d \in \mathcal{D}$ and $x \in \mathsf{X}$, the negative log-probability privacy function is $\rho(d, x) = -\log d(x)$.

In this example, Receiver's decision is a density function. If this density function has a relatively high value at the true data $x$, then Sender has less privacy.

Sender has no explicit control over Receiver's decision $d$, and Receiver's agenda is unknown to Sender. In Section 2.2, we use assumptions to derive Receiver's optimal decision, so that Sender can assess privacy with such a response from Receiver. Before this, we comment on the need for transparent privacy guarantees.

### 2.1.2. TRANSPARENT GUARANTEES

Transparency is important in the context of privacy as Sender is typically required to convince external parties (e.g., regulators or the public) that the mechanism in question satisfies a privacy guarantee. As such, Sender will share the mechanism $M$ and their chosen privacy definition so that the privacy status of $M$ can be verified. Sharing the privacy definition is equivalent to sharing $\mathfrak{C}$, the privacy class generated by the definition (for which $M \in \mathfrak{C}$). We describe adherence to the transparency principle by the following assumption.

**Assumption 1.** Sender shares the mechanism $M$ and privacy class $\mathfrak{C}$, for which $M \in \mathfrak{C}$, with Receiver. Further, the definitions of $M$ and $\mathfrak{C}$ do not depend on the data.

The condition that the data does not determine $M$ ensures no information about $x$ is leaked to Receiver when Sender shares the definition of $M$. For example, consider the constant mechanism $M_x$, defined by $M_x(z, \cdot) = \delta_x$ for all $z \in \mathsf{X}$, where $x$ is the true data. Sharing $M_x$ completely reveals the true data $x$. Assumption 1 explicitly prohibits such dependence of $M$ on the data, so that only dependence through the first argument of $M$ need be considered when constructing privacy definitions (Bailie et al., 2026b). Further, leakage can also occur if the privacy class $\mathfrak{C}$ depends on the data (see the remark in the next section).

## 2.2. Receiver's Decision

To derive Receiver's optimal decision, or best response, we make the following assumption.

**Assumption 2.** Receiver makes Bayesian decisions, i.e., they are *Bayes rational* (Aumann, 1987).

Receiver's uncertainty about the sensitive data is represented by their *belief*, a distribution over $x$. In particular, Receiver holds a prior on the values of the data, a *data-prior* $Q \in \mathcal{P}$, and knows the mechanism $M$ generating the output (by Assumption 1). Receiver updates their belief after observing

the realised output $T$ from $M(x, \cdot)$. With this information Receiver constructs their *data-posterior*,

$$Q_T = Q(\,\cdot \mid T), \tag{1}$$

the Bayes update after observing $T \sim M(x, \cdot)$. Note that the effect of the chosen mechanism $M$ is implicit in $Q_T$. For instance, if, for all $x \in \mathsf{X}$, the probabilities $M(x, \cdot)$ have a common dominating measure $\mu$ and densities $m(x, \cdot)$ with respect to $\mu$, then $Q_T(\mathrm{d}x) \propto Q(\mathrm{d}x)m(x, T)$ where $m(\cdot, T)$ is the likelihood function conditional on $T$. Before continuing to Receiver's optimal decision, we make the following remark about transparency.

*Remark.* Were the privacy class dependent on the data then knowledge of such a class, say $\mathfrak{C}_x$, would restrict the support of the data-posterior to $\mathsf{X}' = \{x \in \mathsf{X} : M \in \mathfrak{C}_x\}$. Therefore, excluding this possibility by Assumption 1 ensures that the data-posterior is specified as in (1), and is not restricted[2] to the set $\mathsf{X}'$.

If Receiver has a loss function $\ell : (\mathcal{D}, \mathsf{X}) \to \mathbb{R}$, then by Assumption 2 Receiver makes Bayes decisions according to $\ell$ and their belief $P \in \mathcal{P}$ about the data. Their optimal decision under $P$ and $\ell$ is therefore $d_\ell^P \in \arg\inf_{d \in \mathcal{D}} \mathbb{E}_{X \sim P}[\ell(d, X)]$, noting that the infimum may not be unique. Specifically, before observing the output $T$, Receiver's belief is their data-prior $Q$, whilst afterward it is their data-posterior $Q_T$.

Interestingly, the worst-case data-averaged loss function (from Sender's point of view) for privacy is $\ell = \rho$, as shown in the proposition below.

**Proposition 1.** *If $\ell = \rho$, then Sender attains the worst-case data-averaged privacy value. That is, $\mathbb{E}_{X \sim P}[\rho(d_\rho^P, X)] \leq \mathbb{E}_{X \sim P}[\rho(d_\ell^P, X)]$ for any $\ell : (\mathcal{D}, \mathsf{X}) \to \mathbb{R}$.*

All proofs are deferred to Appendix E. In light of Proposition 1, we now restrict our attention to the case where Receiver has loss function $\ell = \rho$, where $\rho$ is Sender's privacy function.

**Assumption 3.** Receiver's loss function satisfies $\ell(d, x) = \rho(d, x)$ for all $d \in \mathcal{D}$ and $x \in \mathsf{X}$.

For simplicity's sake, we denote Receiver's decision under Assumption 3 as

$$d^P \in \arg\inf_{d \in \mathcal{D}} \mathbb{E}_{X \sim P}[\rho(d, X)]. \tag{2}$$

---

[2]Conditioning the data-posterior on some additional information is not an inherent problem. Rather, $\mathsf{X}' = \{x \in \mathsf{X} : M \in \mathfrak{C}_x\}$ depends on the privacy definition generating $\mathfrak{C}_x$. The same privacy definition we will come to define using the data-posterior, now conditioned on $\mathsf{X}'$ depending on $\mathfrak{C}_x$. Allowing this possibility would result in a self-referential definition of privacy, hence the condition in Assumption 1.

## 2.3. Sender's Attained Privacy

Sender's attained privacy value under Receiver's optimal decision is $\rho(d^P, x)$, where $P$ is Receiver's belief. After Receiver has observed the output, Sender's privacy value is $\rho(d^{Q_T}, x)$, which will depend on the mechanism chosen. As such, Sender wishes to choose a mechanism which persuades Receiver to make decisions which have a limited impact on privacy.

The attained privacy value also has an interpretation as a proper scoring rule.

**Proposition 2.** *Let $S : (\mathcal{P}, \mathsf{X}) \to \mathbb{R}$ be defined as*

$$S(P, x) = \rho(d^P, x).$$

*Then $S$ is a negatively-oriented proper scoring rule.*

Proposition 2 is proved in Grünwald & Dawid (Section 3.4, 2004) under mild technical conditions (see also, Dawid & Lauritzen, 2005; Dawid, 2007). We call $S$ a *privacy score*, which measures how well Receiver's belief (represented by the distribution $P$) predicts the true value of the data $x$.

Privacy scores can be derived from the privacy functions stated in Examples 1 and 2 as follows.

**Example 1** (continued). The interval privacy score with length $s > 0$ is

$$S(P, x) = \begin{cases} 0 & \text{if } x \in [m^P - \frac{s}{2}, m^P + \frac{s}{2}] \\ 1 & \text{otherwise.} \end{cases}$$

where $m^P = \arg\max_{m \in \mathbb{R}} P([m - \frac{s}{2}, m + \frac{s}{2}])$, assuming the maximiser is unique for all $P \in \mathcal{P}$. When $P$ is a symmetric unimodal distribution, $m^P$ will be its median.

In this example, Receiver's optimal decision is the interval of length $s$ with maximum probability under $P$. If this interval does not contain $x$ then Sender has retained privacy.

**Example 2** (continued). The negative log-probability privacy score is

$$S(P, x) = -\log p(x),$$

where $p$ is the density of $P$ with respect to a given measure $\mu$, in the case where $\mathcal{P}$ consists of distributions which are absolutely continuous with respect to $\mu$.

This example leads to the well-known log-probability score (which is related to the Kullback–Leibler divergence). Appendix A contains a technical note defining the negative log-probability score more generally, required for Section 4.

Proper scoring rules can be constructed from loss functions under Bayes acts (decisions) with mild conditions (Grünwald & Dawid, 2004). As such, working with (proper) privacy scores is equivalent to the privacy-function approach

discussed thus far. That is, the specification of $\rho$ and $\mathcal{D}$ generates a privacy score $S$, whilst a privacy score $S$ implies the existence of an equivalent[3] $\rho$ and $\mathcal{D}$. This allows us to measure privacy using either privacy functions or proper scoring rules with our framework. Proper scoring rules[4] are convenient and natural to consider, so we will focus on these for the remainder of the paper.

## 2.4. Sender's Decision

In this section, we specify how Sender distinguishes between mechanisms, based on Receiver's optimal decision. Our forthcoming assumptions govern Sender's approach to assessing privacy, but we note that the framework established thus far is also amenable to contexts which may necessitate an alternative approach.

We assume Sender assesses mechanisms based on their relative effect on privacy. That is, Sender chooses a mechanism based on its *relative privacy score*

$$\Delta(Q, T, x) = S(Q, x) - S(Q_T, x).$$

The relative privacy score is negatively-orientated for Sender. That is, lower values of $\Delta(Q, T, x)$ represent greater privacy. Typically, $S(Q, x) > S(Q_T, x)$ as privacy decreases after information is released, and $\Delta(Q, T, x)$ will be positive, but given stochasticity in $T$, the relative privacy score can take negative values. An alternative to a relative assessment is for Sender to assess the absolute privacy using $S(Q_T, x)$. However, this would not capture the change in Receiver's belief, which would lead to very strict assessments of privacy. As an extreme example, if Receiver already has full knowledge of the dataset, i.e., $Q = \delta_x$, then every mechanism would have equal absolute privacy.

The relative privacy score is stochastic since it depends on $T \sim M(x, \cdot)$. But Sender must assess privacy before $T$ is generated. The stochasticity can be accounted for in a number of ways, including by expectation or tail probability. We choose the latter to favour robustness in our definition of privacy, whilst Appendix C discusses the former choice in further detail.

**Assumption 4.** For a given data-prior $Q$ and dataset $x$, Sender considers a mechanism $M$ private only if

$$\mathbb{P}_x\left[\Delta(Q, T, x) \leq \kappa\right] \geq 1 - \delta, \tag{3}$$

for some maximum acceptable privacy loss $\kappa \geq 0$ and small probability of failure $0 \leq \delta \ll 1$.

Assumption 4 dictates that Sender wishes to ensure privacy for all events except (possibly) those with low probability.[5] From the game-theoretic perspective, (3) encodes a binary value function (relative privacy under Receiver's optimal decision) for Sender. Further discussion of the binary value function is given in Section 2.5.

So far, we have conditioned on the existence of a known data-prior $Q$. In most cases, it will be difficult for Sender to know Receiver's data-prior, even approximately. For a robust definition of privacy we consider the worst case over a reasonable class of priors $\mathcal{Q}_x \subset \mathcal{P}$ that Receiver may hold. This class may depend on the data value $x$, since this allows us to construct classes where bounds on Receiver's adversarial strength are constant as $x$ varies. We will drop the subscript when $\mathcal{Q}_x$ does not depend on $x$. Sender's consideration of a set of data-priors is formalised as follows.

**Assumption 5.** Sender considers $M$ private if (3) holds uniformly for all data-priors $Q \in \mathcal{Q}_x$ and datasets $x \in \mathsf{X}$.

That (3) holds uniformly over all datasets $x \in \mathsf{X}$ is sufficient to ensure that the privacy class (to be defined) does not depend on the data, as required by Assumption 1.

Assumptions 4 and 5 imply that Sender will assess the privacy of a mechanism $M$ by validating

$$\inf_{x \in \mathsf{X}} \inf_{Q \in \mathcal{Q}_x} \mathbb{P}_x\left[\Delta(Q, T, x) \leq \kappa\right] \geq 1 - \delta. \tag{4}$$

In particular, mechanisms satisfying (4) are robust to (i) the value of the dataset, (ii) the data-prior held by Receiver, and (iii) the (possibly) stochastic outcome of the mechanism.

## 2.5. Game Interpretations

To provide a complete game-theoretic interpretation of our framework, we can take the following view. Suppose a third player, Nature, has a dataset $x \in \mathsf{X}$, where $x$ is unknown to Sender and Receiver. Nature will reveal the dataset to Sender, who will then share information about $x$ to Receiver through a mechanism. Before this, Sender publicly commits to using a mechanism $M(x, \cdot)$ to share information with Receiver. Once $x$ is revealed, Sender shares the output of the mechanism with Receiver.

In this scenario, if Sender wishes to be robust to the data value, data-prior, and randomness of $M$, they can use (4) to assess the mechanism. As such, the inclusion of Nature provides a complete game-theoretic interpretation for protecting all $x \in \mathsf{X}$ (in Assumption 5), whilst Assumption 4 and consideration of the worst $Q \in \mathcal{Q}_x$ (in Assumption 5) can be attributed to Sender's choice to be robust. In this game, if Sender chooses the mechanism from some set $\mathcal{M}$,

---

[3]We prove the latter statement in Appendix B for completeness. Proof of the former statement is in Grünwald & Dawid (2004) with further discussion in Brehmer & Gneiting (2020).

[4]It is also the case that a scoring rule lacking propriety (i.e., not proper) can often be adjusted to gain propriety (Brehmer & Gneiting, 2020).

[5]Considering the expectation of $\Delta(Q, T, x)$, rather than the tail probabilities, can recover Rényi DP for example (see Appendix C).

then their objective function will be

$$v(M) = 1\{M \in \mathfrak{C}\}, \qquad (5)$$

where $\mathfrak{C}$ is the privacy class defined by (4). Therefore, private mechanisms attain unit utility and non-private mechanisms attain zero utility.

Typically, (5) will not have a unique maximum, and the resulting set of "optimal" private mechanisms will be indistinguishable.[6] This characterisation aligns with contemporary privacy definitions, which typically assess if privacy is attained by a given mechanism or not.

In contrast to this game-theoretic interpretation, we make the following comments on real-world assessment of data privacy. Firstly, when Sender is the custodian of the data, we believe that the transparency justification for protecting all $x \in \mathsf{X}$ is more compelling than the use of Nature as a third player. In other words, protecting all $x \in \mathsf{X}$ is better motivated as a sufficient condition for Assumption 1. Secondly, the privacy of a mechanism is typically assessed individually, rather than as a set or utility function, i.e., comparing (4) to (5). Indeed, this is how most, if not all, privacy guarantees are introduced and discussed. Hence, without loss of generality, we focus on privacy assessment of one mechanism for the remainder of the paper.

## 3. Persuasive Privacy

We extend our definition of privacy to include multiple privacy scores (equivalently privacy functions) if required. We denote the (non-empty) set of privacy scores under consideration by $\mathcal{S}$. The class of Receiver data-priors is $\mathcal{Q}_x \subset \mathcal{P}$. The privacy parameters are $\kappa > 0$ which is the maximal allowable change in privacy, and $0 \leq \delta \ll 1$, representing a small probability of failing to meet this maximal change.

**Definition 3** (Persuasive Privacy). A mechanism $M$ is said to be $(\mathcal{S}, \mathcal{Q}_x, \kappa, \delta)$-PP if

$$\inf_{S \in \mathcal{S}} \inf_{x \in \mathsf{X}} \inf_{Q \in \mathcal{Q}_x} \mathbb{P}_x \left[ \Delta_S(Q, T, x) \leq \kappa \right] \geq 1 - \delta, \quad (6)$$

where $\Delta_S(Q, T, x) = S(Q, x) - S(Q_T, x)$.

When $\mathcal{S}$ is a singleton, say $\mathcal{S} = \{S\}$, we use $(S, \mathcal{Q}_x, \kappa, \delta)$ as shorthand for $(\{S\}, \mathcal{Q}_x, \kappa, \delta)$. Though the dependence is not explicitly stated, the choice of mechanism defines the probability $\mathbb{P}_x(\cdot)$ and affects the construction of the data-posterior $Q_T$ in $\Delta_S(Q, T, x)$. We now consider some properties of $(\mathcal{S}, \mathcal{Q}_x, \kappa, \delta)$-PP mechanisms.

---

[6] A natural tie-breaking strategy is to introduce a secondary utility function which selects the "best" private mechanism, in some manner. For example, a function measuring statistical efficiency could be used to select a mechanism from $\mathfrak{C}$.

### 3.1. Composition

We can attain a composition rule for persuasive privacy when the family of posterior distributions $\mathcal{Q}_x$ is closed under Bayes updating by the mechanisms considered. First we define this conjugacy condition, before stating the composition property.

**Definition 4** (Conjugacy). A family of distributions $\mathcal{Q}_x$ is conjugate to a mechanism $M : (\mathsf{X}, \mathcal{T}) \to [0, 1]$ if for every $Q \in \mathcal{Q}_x$ the posterior distribution $Q(\cdot \mid M, T)$ is also in $\mathcal{Q}_x$ almost surely over $T \sim M(z, \cdot)$, for all $z \in \mathsf{X}$.

A composition property for a guarantee specifies the privacy of a composed mechanism, $M_1 \otimes M_2$, where the second mechanism $M_2((x, T), \cdot)$ is possibly dependent on the output of the first $T \sim M_1(x, \cdot)$. We reference measurable spaces $(\mathsf{T}_1, \mathcal{T}_1)$ and $(\mathsf{T}_2, \mathcal{T}_2)$ where $\mathsf{T}_k$ is the output space of $M_k$.

**Proposition 3.** *Let $M_k : (\mathsf{Y}_k, \mathcal{T}_k) \to [0, 1]$ for $k \in \{1, 2\}$ where $\mathsf{Y}_1 = \mathsf{X}$ and $\mathsf{Y}_2 = \mathsf{X} \times \mathsf{T}_1$. Assume $M_1$ is $(\mathcal{S}, \mathcal{Q}_x, \kappa_1, \delta_1)$-PP, and $M_2((\cdot, t), \cdot)$ is $(\mathcal{S}, \mathcal{Q}_x, \kappa_2, \delta_2)$-PP for all $t \in \mathsf{T}_1$. If $\mathcal{Q}_x$ is conjugate to $M_1$, then $M_1 \otimes M_2$ is $(\mathcal{S}, \mathcal{Q}_x, \kappa_1 + \kappa_2, \delta_1 + \delta_2)$-PP.*

### 3.2. Post-Processing

In this section we distinguish between two types of post-processing properties that are desirable privacy guarantees. To assist with the exposition, we use privacy classes generated by privacy definitions. In particular, let D be a specific persuasive privacy guarantee from Definition 3 and denote the privacy class generated by D as $\mathfrak{C}(\mathrm{D})$. Using this description we can describe two types of post-processing properties. We will use a Markov kernel $K : (\mathsf{T}_1, \mathcal{T}_2) \to [0, 1]$. Unlike the case of composition, the kernel $K$ is independent of the data $x$.

**Definition 5** (Receiver Post-Processing). A guarantee D satisfies the receiver post-processing property if $M \in \mathfrak{C}(\mathrm{D})$ implies that $M \otimes K \in \mathfrak{C}(\mathrm{D})$ for all Markov kernels $K$ independent of the data $x$.

The receiver post-processing property dictates that after observing the mechanism output, Receiver cannot gain additional information about the data by post-processing the output. That is, the privacy guarantee remains no matter what transformation is applied to the output by Receiver. The above interpretation of this property is a cornerstone justification for the adversarial robustness of differential privacy. Here, we establish it for persuasive privacy.

**Proposition 4.** *All persuasive privacy guarantees satisfy the receiver post-processing property.*

Next we consider post-processing by Sender. This following alternative is what is known in the literature as the "post-processing inequality".

**Definition 6** (Sender Post-Processing). A guarantee D satisfies the sender post-processing property if $M \in \mathfrak{C}(D)$ implies that $MK \in \mathfrak{C}(D)$ for all Markov kernels $K$ independent of the data $x$.

The mechanism $MK$ is often referred to as "chaining" $M$ and $K$ (Hay et al., 2021). The sender post-processing property dictates that further (random) transformations of a private output by Sender preserves the privacy guarantee, when only the final transformed output is shared with Receiver. This property is useful as a tool to establish privacy for complex mechanisms by transformations of simpler mechanisms for which a guarantee can be established. This is the typical use of the "post-processing inequality" in the literature.

We can also state that receiver post-processing is a special case of sender post-processing by taking $K = \mathrm{Id} \otimes K'$ and observing that $MK = M \otimes K'$, for a Markov kernel $K'$ independent of the data. As such, sender post-processing is a stronger requirement than receiver post-processing.

Whilst persuasive privacy satisfies receiver post-processing, it does not satisfy the sender post-processing property.

**Proposition 5.** *There exists a persuasive privacy guarantee that does not satisfy the sender post-processing property.*

Despite this negative result, it is still trivial for Sender to release output from $MK$ with the same guarantee as $M$. Specifically, releasing output from $M \otimes K$, instead of just from the marginal $MK$, will satisfy the receiver post-processing property but on Sender's side. From the view of Bayesian Persuasion, this result indicates that Sender can control the privacy loss by releasing output from $M$ in addition to $MK$ (jointly), when the influence on decisions exerted by output from $MK$ is worse than $M$ (or not established).

## 4. Differential Privacy

We can interpret some variants of differential privacy in the persuasive privacy framework. First we state the definition for probabilistic differential privacy (PDP, Machanavajjhala et al., 2008; Gotz et al., 2012; Meiser, 2018). To define PDP, we fix a binary neighbour relation[7] "$\sim$". We denote the set of neighbours of some $x \in \mathsf{X}$ by $\mathfrak{N}_x = \{x' \in \mathsf{X} : x' \sim x\}$, and the set of all neighbours by $\mathfrak{N} = \{(x, x') \in \mathsf{X}^2 : x' \in \mathfrak{N}_x\}$. For example, $\mathfrak{N} = \{(x, x') \in \mathsf{X}^2 : H(x, x') \leq 1\}$ where $H$ is the Hamming distance, is typical. We denote a pair of neighbours $(x, x') \in \mathfrak{N}$ with the shorthand $x \sim x'$ for simplicity.

**Definition 7** (Probabilistic Differential Privacy). A mechanism $M$ is said to be $(\varepsilon, \delta)$-PDP if

$$\inf_{x \sim x'} \mathbb{P}_x \left[ m(x, T) \leq \exp\{\varepsilon\} m(x', T) \right] \geq 1 - \delta,$$

where $T \sim M(x, \cdot)$ and $m(x, \cdot)$ is the probability density (mass) function of $M(x, \cdot)$, with respect to a measure that dominates $M(x, \cdot)$ and $M(x', \cdot)$.

We note that alternative, but equivalent, definitions of PDP also exist in the literature (e.g., Meiser, 2018, Definition 4).

Having defined PDP, we now prove an equivalence to PP mechanisms. Let $L$ be the negative log-probability score for discrete distributions,[8] and consider a class of neighbouring alternative hypothesis data-priors $\mathcal{H} = \{Q \in \mathcal{P}_2 : \exists (x, x') \in \mathfrak{N}, Q(\{x, x'\}) = 1\}$, where $\mathcal{P}_2$ is the class of two-component discrete probability distributions. Note that $\mathcal{H}$ does not depend on the value of $x$.

**Proposition 6.** *A mechanism $M$ is $(\varepsilon, \delta)$-PDP if and only if $M$ is $(L, \mathcal{H}, \varepsilon, \delta)$-PP.*

Since pure DP is a special case of PDP, when $\delta = 0$ we recover $\varepsilon$-DP. The proof reveals that the infimum in the definition of persuasive privacy occurs in the limit of Receiver's prior probability on the true data $Q(\{x\}) \to 0$. We can use this to interpret probabilistic differential privacy as protecting against a worst case where Receiver has vanishingly small probability on the truth. That is, under the negative log-probability score and class of neighbouring alternative hypothesis priors, the gain in information about $x$ is largest when Receiver has near zero probability on this outcome.

Proposition 6 can also be proved for a smaller class of data-priors $\mathcal{H}_x = \{Q \in \mathcal{P}_2 : Q(\{x\}) = 1 - Q(\{x'\}) > 0, x' \in \mathfrak{N}_x\}$, limited to the priors which have positive probability on the truth. This indicates that using the larger class gives no additional privacy guarantee. Essentially, we ignore data-priors that have no mass on the truth as there will be no change in the privacy score in this case.

Considering the properties discussed in Section 3.2, it is well known that PDP does not satisfy the post-processing inequality (Kifer & Lin, 2012; Meiser, 2018). However, our discussion reveals that this is not a drawback for the privacy properties of PDP, rather a restriction of the tools that can be used to establish a PDP guarantee. Furthermore, it is trivial to gain a PDP guarantee for a post-processed mechanism using Proposition 4. One can simply augment the output of the transformed mechanism with the output of the original mechanism.

Establishing PDP as a special case of our framework contributes new semantics to the understanding of differential privacy. Sender wishes to limit the relative privacy score,

---

[7]We also use "$\sim$" to relate random variables to their distribution, but the intended meaning will be clear from context.

[8]The negative log-probability score for discrete distributions is formalised in Appendix A.

the difference of negative log-probability scores under the data-posterior and data-prior, and assesses the worst case under the class of neighbouring alternative hypothesis data-priors. Our explicit construction illuminates potential areas of weakness in the differential privacy setup.

A game-theoretic derivation of Rényi DP (Mironov, 2017) is achieved by changing the assessment of privacy from a tail probability condition (Assumption 4) to an expected value condition. The details are provided in Appendix C, along with a connection to the broader notion of $f$-divergence privacy.

## 5. Privacy for Deterministic Mechanisms

In this section we provide two illustrative yet important examples where is it possible to assess the persuasive privacy of a deterministic function of the data. This contrasts differential privacy and its variants, where non-vacuous guarantees cannot be constructed for deterministic mechanisms.

### 5.1. Private Empirical Average

The canonical mechanism considered in data privacy is the empirical average $\bar{x} = \frac{1}{n} \sum_{i=1}^{n} x_i$ for $n \geq 2$. Adding noise to the average can yield a differentially private mechanism. Assuming X is bounded, Laplace or Gaussian noise yield a pure or approximate DP guarantee, respectively (Dwork et al., 2006a). However, in traditional SDC it is often assumed that for large $n$ no noise is actually required for the average to be private (assuming that differencing attacks and group disclosures are not possible, see e.g., Smith & Elliot, 2008), whilst for DP, the scale of the additive noise of a mean goes to zero as $n \to \infty$.

We construct a persuasive privacy guarantee to formalise the intuition that releasing the empirical average of the data with no noise is private, under reasonable assumptions. To begin we set out the constituent elements of this guarantee. For simplicity we assume $X = \mathbb{R}^n$.

The Dawid–Sebastiani Score (DSS, Dawid & Sebastiani, 1999) is a scoring rule depending only on the first two moments of a distribution $P$. The DSS is a proper scoring rule when the variance of $P$ is finite for all $P \in \mathcal{P}$, and can be seen as a special case of the negative log-probability score for Gaussian distributions. For our purposes, we define a marginal version of the DSS.

**Definition 8** (Marginal Dawid–Sebastiani Score)**.** If $Q$ has support on $X \subset \mathbb{R}^n$, the marginal DSS is defined as

$$D_i(Q, x) = \log \sigma_i^2(Q) + \frac{[x_i - \mu_i(Q)]^2}{\sigma_i^2(Q)},$$

where $\mu_i(Q)$ and $\sigma_i^2(Q)$ are the marginal mean and variance, respectively, for the $i$th dimension of $Q$.

The marginal DSS is a proper scoring rule, recovering the $i$th marginal mean and variance. That is, if $D_i(Q, P) = D_i(Q, Q)$ then $\mu_i(Q) = \mu_i(P)$ and $\sigma_i^2(Q) = \sigma_i^2(P)$. From the marginal DSS, we construct a set of scoring rules $\mathcal{I} = \{D_i\}_{i=1}^n$ to measure the privacy of each marginal in the dataset. One interpretation of this choice is that there are $n$ individuals in the dataset each with $x_i \in \mathbb{R}$, and we wish to protect the worst-case outcome for all individuals.

Next we define the class of data-priors held by Receiver. Let $\mathcal{N}(\mu, \Sigma)$ denote a multivariate Gaussian distribution on $\mathbb{R}^n$ for $\mu \in \mathbb{R}^n$ and positive-definite $\Sigma \in \mathbb{R}^{n \times n}$. Let $\Sigma = \sigma^\top \Phi \sigma$ where $\sigma$ is the vector of marginal standard deviations, and $\Phi$ is the correlation matrix of $\Sigma$. Finally, let $c_\Phi \in [1, \infty)$ denote the condition number of $\Phi$ and define the averages $\bar{\mu} = \frac{1}{n} \sum_{i=1}^n \mu_i$ and $\overline{\Sigma} = \frac{1}{n^2} \sum_{i=1}^n \sum_{j=1}^n [\Sigma]_{ij}$. Note that $\bar{\mu}$ and $\overline{\Sigma}$ are the mean and variance of $\overline{X}$, respectively, where $X \sim \mathcal{N}(\mu, \Sigma)$. Define the class of data-priors $\mathcal{G}_x^r$ as the set

$$\left\{ \mathcal{N}(\mu, \Sigma) : \frac{(\bar{x} - \bar{\mu})^2}{\overline{\Sigma}} \leq r_1, c_\Phi \leq r_2 \left( 1 - \frac{\sigma_i^2}{\|\sigma\|_2^2} \right) \forall i \right\},$$

for $r_1 > 0$ and $r_2 > 1$. The first condition defining $\mathcal{G}_x^r$ states that Receiver's prior guess for $\bar{x}$ cannot be too poor, relative to the variance of the average under Receiver's prior. In contrast, the second condition limits how strong Receiver's prior is, by controlling degeneracy in the data-prior.

**Proposition 7.** *The average mechanism $M(x, \cdot) = \delta_{\bar{x}}$ satisfies $(\mathcal{I}, \mathcal{G}_x^r, \kappa^r, 0)$-PP with $\kappa^r = r_1 + \log r_2$.*

Proposition 7 establishes a persuasive privacy guarantee for a deterministic mechanism. The parameters $(\kappa^r, 0)$ of the privacy guarantee are determined by the relative strength of Receiver, which is specified by the values $r_1$ and $r_2$ defining the class of data-priors $\mathcal{G}_x^r$.

The data-posterior that is considered in this example is a degenerate Gaussian distribution constrained to the manifold determined by $\bar{x}$. In some sense, we avoid the degeneracy and complexity of the multivariate data-posterior by using the marginal DSS to assess privacy. This is an interesting contrast to the persuasive privacy interpretation of PDP, where the complexity of the multivariate distribution is handled by using a restrictive class of discrete data-priors.

In light of Proposition 7, the data-prior restrictions in $\mathcal{G}_x^r$ have the following interpretation. Firstly, Receiver gains too much information when they hold a data-prior where the average of the means, $\bar{\mu}$, is far away from the truth with strong conviction (i.e., a small $\overline{\Sigma}$ value). Secondly, no single marginal variance can account for too much of the total marginal variance, relative to the condition number of the correlation matrix, which measures the overall strength of correlations. Otherwise, revealing $\bar{x}$, which constrains the prior to an $(n-1)$-dimensional manifold, will indirectly

reveal too much about the $i$th component (assuming $r_1$ is small). This can occur when $\sigma_i \gg \sigma_j$ for all $j \neq i$, since knowledge of the manifold and the remaining components ($j \neq i$) with high certainty determines $x_i$.

Clearly, $(\mathcal{I}, \mathcal{G}^r_x, \kappa^r, 0)$-PP is not robust to all Receivers with Gaussian priors, as we restrict the class of priors considered. However, it may be possible for robust versions of the mean (or indeed other deterministic mechanisms) to have such a property. Considering the interactions between stochasticity and robustness under our general framework is left for future work.

### 5.2. Private Cell Suppression

In this example, Sender wishes to safely reveal a vector of counts $x \in \mathbb{N}^n$ by suppressing entries with small values. The vector $x$ may relate to a histogram or contingency table summarising subpopulation counts for example, cases that frequently arise in official statistics where cell suppression is a common disclosure control method (Australian Bureau of Statistics, 2021; U.S. Centers for Disease Control and Prevention, 2026). We will refer to each element $x_i$ as a cell and assume Sender assesses privacy individually for each cell. Suppose Sender considers any cell value greater than $x_i \geq r$ for some $r \geq 2$ to be inherently private. Intuitively, Sender is stating that a count of $r$ in a cell is enough to obscure any information about a single individual. In contrast, for cells with $x_i < r$ Sender uses the (marginal) log-probability score to assess privacy. Hence, the privacy score for the $i$th cell can be written as

$$L_i^r(Q, x) = \begin{cases} -\log q_i(x_i) & \text{if } x_i < r, \\ \infty & x_i \geq r, \end{cases}$$

where $q_i$ is the $i$th marginal PMF of the distribution $Q$.

We assume that, under Receiver's data-prior, the distribution of each cell belongs to the set $\mathcal{C}^{h,\alpha} = \{P \in \mathcal{P}_{\mathbb{N}} : P(\{z < h\}) \geq \alpha\}$, where $\mathcal{P}_{\mathbb{N}}$ is the set of distributions over the natural numbers. Further, the cells are independent, so that the Receiver's data priors are the $n$-product of $\mathcal{C}^{h,\alpha}$ which we denote by $\mathcal{C}_n^{h,\alpha}$. Here $h \geq 2$ is the suppression threshold and $\alpha$ expresses the strength of Receiver's belief in $x_i < h$.

The cell-suppression mechanism releases the value of $x_i$ deterministically if $x_i \geq h$, and otherwise censors the cell. That is, for each cell $i$, we have $M_i^h(x_i, \{x_i\}) = 1$ if $x_i \geq h$, and $M_i^h(x_i, \{-1\}) = 1$ otherwise, where $-1$ denotes suppression. We now state the privacy guarantee.

**Proposition 8.** *If $h \geq r$, the cell-suppression mechanism $\prod_{i=1}^n M_i^h(x_i, \cdot)$ satisfies $(\mathcal{L}_n^r, \mathcal{C}_n^{h,\alpha}, -\log \alpha, 0)$-PP.*

In this setting, $\alpha$ is a lower bound for the weakness of Receiver. Intuitively, a weak Receiver has a data-prior with low probability on small cells counts, and learns more (than a strong Receiver) when a cell is suppressed. To protect against weak Receivers, Sender can increase the suppression threshold $h$ to lower the corresponding $\alpha$.

This example demonstrates another deterministic mechanism for which a persuasive privacy guarantee can be instantiated. Whilst the class of Receiver data-priors considered is large, the independence assumption is strong. Future work could relax this by considering cell summation constraints or parametric prior classes with dependence.

## 6. Conclusion

We have considered a class of robust asymmetric Stackelberg games to construct a framework for defining privacy guarantees. In this game, Sender can be thought to model Receiver as a Bayesian agent, whilst making decisions that are robust to stochasticity and the data-prior Receiver may hold. This setup has several potential benefits, including the possibility of generating fit-for-purpose guarantees using privacy functions and scores. We demonstrated how to recover PDP using our framework and also accommodated Rényi DP with a simple modification of our assumptions. Beyond this we focussed on examples of privacy guarantees for deterministic mechanisms, but acknowledge that we are yet to systematically demonstrate how scoring rules can be used to adapt privacy guarantees to different contexts (see in particular the literature on the difficulties in valuing privacy in real-world situations—e.g., Acquisti et al., 2016; Lindgreen, 2018). This direction is left for future work.

The ability to construct persuasive privacy guarantees for deterministic mechanisms suggests the potential for new privacy definitions with improved privacy–utility trade-offs. However, care must be taken to ensure that such guarantees are adequate for their specific context. For example, the privacy guarantee for the empirical average in Section 5.1 achieves perfect statistical efficiency, at the cost of a restricted class of data-priors and the specific use of the marginal DSS as a privacy score.

More generally, the flexibility in persuasive privacy definitions can be misused. Privacy scores or data-priors can be chosen that are inappropriate for the data context. However, the transparent communication of a privacy guarantee means that a given guarantee can always be tested for appropriateness where it is deployed. Practically, the class of priors should be chosen to be as large as possible whilst remaining realistic for the given scenario.

We did not explicitly consider statistical utility in our work, but general methods for determining the utility-optimal mechanisms in a given privacy class is an interesting future research direction. Further, work applying our framework to specific contexts whilst reasoning about appropriate data-prior classes and privacy scores is ongoing.

## Acknowledgements

We are grateful to participants of the Les Houches Privacy Workshop in 2024 and 2025 for lively discussions that informed this paper, with special thanks to Andrea Bertazzi and Stanislas du Ché. Further thanks to participants at the Venice Privacy Workshop in 2026. We also acknowledge the four anonymous reviewers for their helpful comments. JJB was supported by a 2025 Early Mid-Career Research Grant from the University of Adelaide. JJB and CPR were supported by the European Union (ERC-2022-SYGOCEAN-101071601). Views and opinions expressed are however those of the author(s) only and do not necessarily reflect those of the European Union or the European Research Council Executive Agency. Neither the European Union nor the granting authority can be held responsible for them. CPR is further funded by a PR[AI]RIE-PSAI chair from the Agence Nationale de la Recherche (ANR-23-IACL-0008).

## Impact Statement

This paper presents work whose goal is to advance the field of machine learning. There are many potential societal consequences of our work, none of which we feel must be specifically highlighted here.

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

# A. The Negative Log-Probability Score for Discrete Distributions

To accommodate the set of all discrete distributions (which does not have a common dominating measure) we define the negative log-probability score for discrete distributions as follows. Let $(\mathsf{X}, \mathcal{X})$ be a measurable space such that $\{x\} \in \mathcal{X}$ for all $x \in \mathsf{X}$. Consider probability measures of the form $Q = \sum_{z \in \mathsf{Z}} w_z \delta_z$ for some countable set $\mathsf{Z} \subset \mathsf{X}$, with $w_z > 0$ for all $z \in \mathsf{Z}$ and $\sum_{z \in \mathsf{Z}} w_z = 1$. We denote the set of probability measures of this form by $\mathcal{P}_{\leq \aleph_0}$.

**Definition 9.** Let the discrete negative log-probability score $L : (\mathcal{P}_{\leq \aleph_0}, \mathsf{X}) \to \mathbb{R} \cup \{-\infty, \infty\}$ be defined as

$$L(Q, x) = -\log Q(\{x\}),$$

with convention that $\log 0 = -\infty$.

Hence, for any $Q = \sum_{z \in \mathsf{Z}} w_z \delta_z \in \mathcal{P}_{\leq \aleph_0}$, the score is $L(Q, z) = -\log w_z$ for $z \in \mathsf{Z}$ and $L(Q, z) = \infty$ for $z \notin \mathsf{Z}$. The scoring rule $L$ has the following property.

**Proposition 9.** *The scoring rule $L$ is strictly proper relative to the class of distributions $\mathcal{P}_{\leq \aleph_0}$.*

*Proof.* Consider $P, Q \in \mathcal{P}_{\leq \aleph_0}$. We have $P = \sum_{y \in \mathsf{Y}} v_y \delta_y$ and $Q = \sum_{z \in \mathsf{Z}} w_z \delta_z$ for some $\mathsf{Y}, \mathsf{Z} \subset \mathsf{X}$ and positive probabilities $v_y$ and $w_z$ defined over $y \in \mathsf{Y}$ and $z \in \mathsf{Z}$, respectively. The expected score $L(P, Q)$ is

$$L(P, Q) = \begin{cases} -\sum_{z \in \mathsf{Z}} w_z \log v_z & \text{if } \mathsf{Z} \subset \mathsf{Y}, \\ \infty & \text{otherwise}, \end{cases}$$

whilst $L(Q, Q) = -\sum_{z \in \mathsf{Z}} w_z \log w_z$.

Case 1. If $\mathsf{Z} \not\subset \mathsf{Y}$ then $L(Q, Q) < L(P, Q) = \infty$ and clearly $P \neq Q$.

Case 2. If $\mathsf{Z} \subset \mathsf{Y}$ then $L(P, Q) = -\sum_{z \in \mathsf{Z}} w_z \log v_z \geq -\sum_{z \in \mathsf{Z}} w_z \log v_z'$ where $v_z' = \frac{v_z}{\sum_{u \in \mathsf{Z}} v_u} \geq v_z$ are renormalised probabilities. Hence $L(P, Q) \geq -\sum_{z \in \mathsf{Z}} w_z \log w_z = L(Q, Q)$ by Gibbs' inequality, where equality holds if and only if $v_z' = w_z$ for all $z \in \mathsf{Z}$. Note that in the case of equality $\mathsf{Z} = \mathsf{Y}$ and $v_z' = v_z$ for all $z \in \mathsf{Z}$ due to the positive probability constraints on the weights, and then $-\sum_{z \in \mathsf{Z}} w_z \log v_z = -\sum_{z \in \mathsf{Z}} w_z \log w_z$.

Therefore, $L(Q, Q) \leq L(P, Q)$ for all $P, Q \in \mathcal{P}_{\leq \aleph_0}$ and $L(P, Q) = L(Q, Q)$ if and only if $P = Q$. $\square$

We can also conclude that the discrete negative log-probability score is strictly proper with respect to $\mathcal{P}_2$, the class of discrete two-component probability distributions, i.e., the class of $Q = \sum_{z \in \mathsf{Z}} w_z \delta_z$ with $|\mathsf{Z}| = 2$.

# B. The Existence of a Decision Problem for Any Proper Scoring Rule

Consider a probability distribution $P \in \mathcal{P}$ on the measurable space $(\mathsf{X}, \mathcal{X})$. We say that a Bayesian decision problem, a tuple $(\ell, \mathcal{D})$ with loss function $\ell$ and decision space $\mathcal{D}$, generates the scoring rule $S(P, x) = \ell(d^P, x)$ for $P \in \mathcal{P}$ and $x \in \mathsf{X}$, where $d^P = \arg \inf_{d \in \mathcal{D}} \mathbb{E}_{X \sim P}[\ell(d, X)]$.

The following proposition demonstrates that every proper scoring rule $S$ has an associated Bayesian decision problem that generates $S$. Let $S : (\mathcal{P}, \mathsf{X}) \to \mathbb{R} \cup \{-\infty, \infty\}$ be a scoring rule.

**Proposition 10.** *If $S$ is proper with respect to $\mathcal{P}$ then $\ell : (\mathcal{P}, \mathsf{X}) \to \mathbb{R} \cup \{-\infty, \infty\}$ defined as*

$$\ell(d, x) = S(d, x),$$

*defines a Bayesian decision problem $(\ell, \mathcal{P})$ that generates $S$ as the corresponding scoring rule.*

The proof is somewhat trivial but included for completeness.

*Proof.* Let $Q \in \mathcal{P}$. If $\ell(d, x) = S(d, x)$ then $\mathbb{E}_{X \sim Q}[\ell(d, X)] = \mathbb{E}_{X \sim Q}[S(d, X)] = S(d, Q)$ is minimised at $d^Q = Q$ since $S$ is proper. Let $S'(Q, x)$ be the proper scoring rule generated by the decision problem $(\ell, \mathcal{P})$. Since $d^Q = Q$, the scoring rule $S'$ satisfies $S'(Q, x) = \ell(d^Q, x) = S(Q, x)$, as required. $\square$

## C. Rényi Differential Privacy, $f$-Divergence Privacy and Privacy in Expectation

In the main text, Assumption 4 dictates that a persuasive privacy definition is a tail probability condition. In this section, we instead assume that privacy is assessed with an expectation condition. That is, we replace Assumption 4 with the following alternative.

**Assumption 4\*.** *For a given data-prior $Q$ and dataset $x$, Sender considers a mechanism $M$ private only if*

$$\mathbb{E}_x\left[g\left\{\Delta(Q,T,x)\right\}\right)\right] \leq \kappa, \tag{7}$$

*for some maximum acceptable privacy loss $\kappa \geq 0$ and suitable transformation $g : \mathbb{R} \to \mathbb{R}$.*

In (7) the expectation is computed with respect to $T \sim M(x,\cdot)$. We will use $\mathbb{E}_x$ as shorthand notation for the remainder of this section. The choice of $g$ must be considered carefully to ensure that privacy statements are not vacuous for the chosen relative privacy score. Intuitively, at least, we expect $g$ to be non-decreasing to preserve the orientation of the relative privacy score. Further, note that Assumption 5 is unchanged,[9] but now refers to (7) rather than (3). Thus, we can define *Persuasive Privacy in Expectation* (PPE) as follows.

**Definition 10** (Persuasive Privacy in Expectation). A mechanism $M$ is said to be $(\mathcal{S}, \mathcal{Q}_x, g, \kappa)$-PPE if

$$\sup_{S \in \mathcal{S}} \sup_{x \in \mathsf{X}} \sup_{Q \in \mathcal{Q}_x} \mathbb{E}_x\left[g\{\Delta_S(Q,T,x)\}\right] \leq \kappa. \tag{8}$$

Unlike the case for persuasive privacy, where we want the worst-case relative privacy score to be bounded (with high probability), here we control the worst-case expected relative privacy score. This explains the use of supremum in (8), rather than infimum as in (6).

We will now show that Rényi DP (Mironov, 2017), among a wider class of privacy definitions (including $f$-divergence privacy), can be recovered from Definition 10. Recall that, for $\alpha \in [1, \infty)$, the $\alpha$-Rényi divergence $D_\alpha$ is defined as

$$D_\alpha[P\|P'] = \begin{cases} \frac{1}{\alpha-1}\log\mathbb{E}_{X\sim P}\left[\frac{\mathrm{d}P}{\mathrm{d}P'}(X)^{\alpha-1}\right] = \frac{1}{\alpha-1}\log\mathbb{E}_{X\sim P'}\left[\frac{\mathrm{d}P}{\mathrm{d}P'}(X)^{\alpha}\right] & \text{if } \alpha > 1, \\ \mathbb{E}_{X\sim P}\left[\log\frac{\mathrm{d}P}{\mathrm{d}P'}(X)\right] = \mathbb{E}_{X\sim P'}\left[\frac{\mathrm{d}P}{\mathrm{d}P'}(X)\log\frac{\mathrm{d}P}{\mathrm{d}P'}(X)\right] & \text{if } \alpha = 1, \end{cases} \tag{9}$$

where $\mathbb{E}_P$ denotes the expectation with respect to $P$. The case $\alpha = 1$ corresponds to the Kullback–Leibler divergence. $\alpha$-Rényi divergence can also be defined for $\alpha = \infty$ by taking the limit of $D_\alpha$ as $\alpha \to \infty$. The resulting privacy definition coincides with pure $\varepsilon$-DP and is omitted here.

Using Rényi divergences we can define Rényi DP as follows.

**Definition 11** (Rényi Differential Privacy). A mechanism $M$ is said to be $(\alpha, \varepsilon)$-RDP for some $\alpha \geq 1$ and $\varepsilon \geq 0$ if

$$\sup_{x \sim x'} D_\alpha[M(x',\cdot)\|M(x,\cdot)] \leq \varepsilon.$$

Note the symmetry in $x$ and $x'$ to obtain the original expression in Mironov (2017). We state below two propositions relating Rényi DP definitions to an equivalent PPE guarantee.

Recall the class of neighbouring alternative hypothesis data-priors $\mathcal{H} = \{Q \in \mathcal{P}_2 : \exists z \sim z', Q(\{z, z'\}) = 1\}$ and consider log-probability score $L(Q, x) = -\log Q(\{x\})$ for $Q \in \mathcal{H}$ as in Appendix A. Further, let $\mathrm{id} : \mathbb{R} \to \mathbb{R}$ denote the identity function, $\mathrm{id}(x) = x$ for $x \in \mathbb{R}$.

**Proposition 11.** *A mechanism $M$ is $(1, \varepsilon)$-RDP if and only if $M$ is $(L, \mathcal{H}, \mathrm{id}, \varepsilon)$-PPE.*

*Proof.* If $M$ is $(L, \mathcal{H}, \mathrm{id}, \varepsilon)$-PPE then

$$\sup_{x \in \mathsf{X}} \sup_{Q \in \mathcal{H}} \mathbb{E}_x\left[\Delta_L(Q,T,x)\right] \leq \varepsilon,$$

where $\Delta_L(Q, T, x) = \log\frac{m(x,T)}{m(x,T)w + m(x',T)(1-w)}$ for some $w \in (0,1)$ if $Q(\{x\}) > 0$ and zero otherwise, following Section E.5, with $m(x,\cdot)$ and $m(x',\cdot)$ respectively the densities of $M(x,\cdot)$ and $M(x',\cdot)$ with respect to a dominating

---

[9]Note that alternatives to Assumption 5 using expectations could also be explored.

measure (which may depend on $x$ and $x'$). When $Q(\{x\}) = 0$, $\mathbb{E}_x[\Delta_L(Q,T,x)] = 0$, and otherwise

$$\mathbb{E}_x[\Delta_L(Q,T,x)] = \mathbb{E}_x\left[\log\left(\frac{m(x,T)}{wm(x,T)+(1-w)m(x',T)}\right)\right],$$

which is convex in $w$, so that its supremum over $w \in (0,1)$ is

$$\max\left\{\mathbb{E}_x\left[\log\frac{m(x,T)}{m(x',T)}\right],\ \mathbb{E}_x\left[\log\frac{m(x,T)}{m(x,T)}\right]\right\} = \mathbb{E}_x\left[\log\frac{m(x,T)}{m(x',T)}\right],$$

and hence

$$\sup_{x\in\mathsf{X}}\sup_{Q\in\mathcal{H}}\mathbb{E}_x\left[g\{\Delta_L(Q,T,x)\}\right] = \sup_{x\sim x'}D_1[M(x,\cdot)\|M(x',\cdot)],$$

as required. $\qquad\square$

**Proposition 12.** *For $\alpha > 1$, a mechanism $M$ is $(\alpha,\varepsilon)$-RDP if and only if $M$ is $(L,\mathcal{H},g_\alpha,e^{(\alpha-1)\varepsilon})$-PPE where $g_\alpha(s) = e^{(\alpha-1)s}$.*

*Proof.* If $M$ is $(L,\mathcal{H},g_\alpha,e^{(\alpha-1)\varepsilon})$-PPE then

$$\sup_{x\in\mathsf{X}}\sup_{Q\in\mathcal{H}}\mathbb{E}_x\left[g_\alpha\{\Delta_L(Q,T,x)\}\right] \le e^{(\alpha-1)\varepsilon}, \qquad (10)$$

where $\Delta_L(Q,T,x) = \log\frac{m(x,T)}{m(x,T)w+m(x',T)(1-w)}$ for some $w \in (0,1)$ if $Q(\{x\}) > 0$ and zero otherwise, following Section E.5, with $m(x,\cdot)$ and $m(x',\cdot)$ respectively the densities of $M(x,\cdot)$ and $M(x',\cdot)$ with respect to a dominating measure (which may depend on $x$ and $x'$). When $Q(\{x\}) = 0$, $\mathbb{E}_x[g_\alpha\{\Delta_L(Q,T,x)\}] = 1$, and otherwise

$$\mathbb{E}_x[g_\alpha\{\Delta_L(Q,T,x)\}] = \mathbb{E}_x\left[\left(\frac{m(x,T)}{wm(x,T)+(1-w)m(x',T)}\right)^{\alpha-1}\right],$$

which is convex in $w$ so that its supremum in $w \in (0,1)$ is equal to

$$\max\left\{1,\ \mathbb{E}_x\left[\left(\frac{m(x,T)}{m(x',T)}\right)^{\alpha-1}\right]\right\} = \exp\left\{(\alpha-1)D_\alpha[M(x,\cdot)\|M(x',\cdot)]\right\},$$

and hence

$$\sup_{x\in\mathsf{X}}\sup_{Q\in\mathcal{H}}\mathbb{E}_x\left[g_\alpha\{\Delta_L(Q,T,x)\}\right] \le e^{(\alpha-1)\varepsilon} \quad\Longleftrightarrow\quad \sup_{x\sim x'}D_\alpha[M(x,\cdot)\|M(x',\cdot)] \le \varepsilon. \qquad\square$$

We now establish an equivalence between PPE and a class of privacy guarantees generated by $f$-divergences.

Let $f : [0,\infty) \to (-\infty,\infty]$ be a convex function satisfying $f(1) = 0$, $f(s) < \infty$ for $s > 0$ and $f(0) = \lim_{t\to 0^+} f(t)$ (with the possibility that this limit is infinite). We will say that any function $f$ satisfying the above requirements is *suitable* to define an $f$-divergence. We recall that the $f$-divergence $D_f$ induced by $f$ is defined as

$$D_f[P\|P'] = \mathbb{E}_{X\sim P'}\left[f\left(\frac{\mathrm{d}P_a}{\mathrm{d}P'}(X)\right)\right] + f'(\infty)P_s(\mathsf{T}), \qquad (11)$$

for two probabilities $P$ and $P'$ defined on the same space $(\mathsf{T},\mathcal{T})$, where $P = P_a + P_s$ is the Lebesgue decomposition of $P$ with respect to $P'$ and $f'(\infty) := \lim_{t\to\infty} f(t)/t$.

The following proposition can be seen as a generalisation of the previous results for Rényi DP (since $\alpha$-Rényi divergences are monotone transformations of $f$-divergences).

**Proposition 13.** *Suppose $f$ is a suitable function for defining an $f$-divergence $D_f$, and let $g(s) = f(\exp\{-s\})$ for $s \in \mathbb{R} \cup \{\infty\}$. If $f$ is lower semicontinuous, then a mechanism $M$ satisfies*

$$\sup_{x\sim x'}D_f[M(x',\cdot)\|M(x,\cdot)] \le \kappa, \qquad (12)$$

*if and only if $M$ satisfies $(L,\mathcal{H},g,\kappa)$-PPE.*

Equation 12 is a variant of differential privacy called $f$-*divergence privacy* (Barber & Duchi, 2014; Barthe & Olmedo, 2013).

*Proof.* Recall that $f : [0, \infty) \to (-\infty, \infty]$ with $f(1) = 0$. If $M$ is $(L, \mathcal{H}, g, \kappa)$-PPE then

$$\sup_{x \in \mathsf{X}} \sup_{Q \in \mathcal{H}} \mathbb{E}_x \left[ g\{\Delta_L(Q, T, x)\} \right] \leq \kappa,$$

where $\Delta_L(Q, T, x) = \log \frac{m(x,T)}{m(x,T)w + m(x',T)(1-w)}$ for some $w \in (0, 1)$ if $Q(\{x\}) > 0$ and zero otherwise, following Section E.5, with $m(x, \cdot)$ and $m(x', \cdot)$ respectively the densities of $M(x, \cdot), M(x', \cdot)$ with respect to a dominating measure, say $\mu$ (which may depend on $x$ and $x'$). When $Q(\{x\}) = 0$, $\mathbb{E}_x[g\{\Delta_L(Q, T, x)\}] = 0$, and otherwise

$$\mathbb{E}_x[g\{\Delta_L(Q, T, x)\}] = \int f\left( \frac{m(x,t)w + m(x',t)(1-w)}{m(x,t)} \right) m(x,t)\mu(\mathrm{d}t)$$
$$= D_f[wM(x, \cdot) + (1-w)M(x', \cdot) \| M(x, \cdot)],$$

where the second equality follows from the fact that, for $a > 0$, we define

$$f\left(\frac{a}{0}\right) \times 0 := \lim_{b \to 0^+} bf\left(\frac{a}{b}\right) = af'(\infty).$$

Using the joint convexity property of $f$-divergences, we obtain

$$D_f[wM(x, \cdot) + (1-w)M(x', \cdot) \| M(x, \cdot)] \leq (1-w)D_f[M(x', \cdot) \| M(x, \cdot)] \leq D_f[M(x', \cdot) \| M(x, \cdot)],$$

where the equality is attained at $w = 0$. Hence

$$\sup_{w \in (0,1)} D_f[wM(x, \cdot) + (1-w)M(x', \cdot) \| M(x, \cdot)] \leq D_f[M(x', \cdot) \| M(x, \cdot)],$$

and moreover,

$$\liminf_{w \to 0^+} D_f[wM(x, \cdot) + (1-w)M(x', \cdot) \| M(x, \cdot)] \geq D_f[M(x', \cdot) \| M(x, \cdot)],$$

by lower semicontinuity of $f$ (Ambrosio et al., 2000, Theorem 2.34). Together, these two inequalities imply

$$\sup_{w \in (0,1)} D_f[wM(x, \cdot) + (1-w)M(x', \cdot) \| M(x, \cdot)] = D_f[M(x', \cdot) \| M(x, \cdot)].$$

As such, we have

$$\sup_{x \in \mathsf{X}} \sup_{Q \in \mathcal{H}} \mathbb{E}_x \left[ g\{\Delta_S(Q, T, x)\} \right] = \sup_{x \sim x'} D_f[M(x', \cdot) \| M(x, \cdot)]. \qquad \square$$

*Remark.* Choosing an $f$ that is non-increasing is required to ensure $g$ is non-decreasing, for the sake of suitability of $g$ in (7), but is not required for the proof.

# D. Persuasive Privacy's Relation to Existing Privacy Definitions

### D.1. Quantitative Information Flow

Quantitative information flow (QIF) is a framework for quantifying the amount of secret information leaked by a data release (Alvim et al., 2012; 2020; Smith, 2009). The basic intuition behind QIF is to measure how much uncertainty in a secret is reduced after observing the output of the data release mechanism $M$. Here, a secret should be understood as some aspect of the confidential data $x$ that is inputted into the mechanism $M$. More technically, a secret is simply a random variable that, from Receiver's point of view, has a joint distribution with the output of the mechanism. The loss in uncertainty (or dually, the gain in information) in QIF could be measured by the difference between the prior and posterior Shannon entropy of the secret, for example. This basic intuition behind QIF has parallels with persuasive privacy definitions for which Receiver's loss is a measure of uncertainty of their belief in the secret.

There are further similarities between QIF and persuasive privacy. Both are measures of the data release mechanism, rather than measures of the observed output, and they both assume transparency of the privacy guarantee and the mechanism. Like persuasive privacy, QIF uses Bayesian agents. They both define privacy in terms of an attacker's loss (or dually, an attacker's

"gain" in QIF terminology), under the assumptions that the attacker is Bayesian and that they take their Bayes optimal action. Moreover, they likewise view this privacy loss as relative—i.e., they both compare loss before and after the mechanism's release.

However, there are two substantial differences between QIF and persuasive privacy. In QIF, the absolute privacy loss is an average of the optimal losses under $Q_T$, averaged over a (marginal) distribution of $T$. That is to say, the privacy loss in QIF equals $\mathbb{E}_{Z \sim Q} \mathbb{E}_{T \sim M(Z, \cdot)}[\inf_{d \in D} \mathbb{E}_{X \sim Q_T}[\ell(d, X)]]$ for a fixed prior $Q$. This loss can also be derived in our framework by using Assumption 4* (as in persuasive privacy in expectation, see Appendix C) and changing Assumption 5 to consider the expected value of the data under the $Q$, rather than a worst-case analysis. From a game perspective this assumes that Sender and Receiver share the same data-prior, and Sender will assess privacy according to this data-prior.

As proponents of QIF discuss (Boreale & Paolini, 2015; Boreale, 2013), this averaging makes QIF potentially inadequate for assessing information leakage. One approach to rectify this would be to average first over the conditional distribution of $T$ given $x$, and then consider the worst-case over $x$—this is exactly the approach taken in our formulation of persuasive privacy in expectation (see Appendix C). In comparison, the approach taken by persuasive privacy is to look at the tail probabilities of the (unaveraged) relative privacy loss. We view this as more robust than QIF's approach to privacy. It provides stronger guarantees, because we ensure worst-case relative privacy loss is bounded (while allowing for the possibility of a small $\delta$ failure probability). It should be noted, however, that our framework and QIF likewise allow for robustness in two other ways, by requiring that a privacy guarantee holds over both (i) multiple different data priors; and (ii) multiple different privacy loss functions.

### D.2. Differential Privacy

Like QIF and persuasive privacy, differential privacy (DP) is a family of technical definitions that measure the privacy of a data release mechanism. DP definitions are unified by their formulation of privacy as the rate of change—i.e., the 'derivative'—of the mechanism (echoing the epithet 'differential'). While they differ on how they define this derivative, they all follow the same basic idea: measure the change in the (distribution of) the mechanism's output due to counterfactual changes in the mechanism's input (Bailie et al., 2026a;b). This differs from the present paper's view of privacy, which does not consider pairs of counterfactual input datasets $(x, x')$ but takes the worst-case privacy loss over all possible $x$. Yet, we still recover probabilistic $(\varepsilon, \delta)$-DP (for which pure $\varepsilon$-DP is a special case) through the negative log-probability score and two-component discrete data-priors (Proposition 6).

Moreover, Appendix C modifies a key assumption of persuasive privacy, leading to a new framework—persuasive privacy in expectation—which evaluates privacy in terms of expectations rather than tail probabilities. Under this framework, we recover Rényi DP and $f$-divergence privacy (Propositions 11–13). Therefore, our work subsumes many of the common flavours of DP.[10]

We compare persuasive privacy with another variant of DP—pufferfish privacy, which, like persuasive privacy, is Bayesian—in the following section (Appendix D.3). We expect that there are further connections with other flavours of DP. For example, relating capacity-bounded DP (Chaudhuri et al., 2019) to persuasive privacy may be possible, since it is a generalisation of Rényi DP. Besides $(\varepsilon, \delta)$-DP and Rényi DP, there are other common flavours of DP—in particular, Gaussian DP, $f$-DP more generally (Dong et al., 2022), and zero-concentrated DP (Bun & Steinke, 2016). Deriving equivalent persuasive privacy definitions to these DP variants will provide insights into what privacy protections these flavours actually provide. However, we leave these investigations to future work.

Nevertheless, the recovery of Rényi DP in our framework demonstrates the relation between $(1, \varepsilon)$-RDP and $\varepsilon$-DP with our semantics-first approach. Both definitions can be recovered with our game-theoretic setup under almost identical settings. In particular, the choice of Assumption 4 or Assumption 4* separates the definitions; DP assesses privacy with a tail probability whereas RDP assesses privacy in expectation. In contrast, Mironov (2017) showed $\varepsilon$-DP can be recovered as $\infty$-RDP, the limiting case as $\alpha \to \infty$.

### D.3. Pufferfish Privacy and Its Variants, Including Noiseless Privacy

Pufferfish privacy is a type of DP which incorporates attackers' uncertainty in the sensitive dataset $x$ (Kifer & Machanava-jjhala, 2014). It models this uncertainty using a Bayesian formalism by placing priors on $x$. With these priors, pufferfish

---

[10]Note that our definitions are all agnostic to how neighbouring datasets are defined; that is to say, our definitions cover probabilistic DP, pure DP, Rényi DP and $f$-divergence privacy regardless of the choice of neighbours.

privacy considers a different type of "data release mechanism" $M_{\text{Ext}}$ that is constructed from any standard mechanism $M$: Instead of taking as input the sensitive dataset $x$, $M_{\text{Ext}}$ takes as input a data-prior $Q$, the dataset $x$ is then generated according to $Q$ and, lastly, $x$ is fed into $M$ to produce the output. That is to say, $M_{\text{Ext}}(Q)$ outputs $T$ drawn from the prior predictive distribution $\mathbb{P}_Q(T \in \cdot) := \int_{\mathsf{X}} \mathbb{P}_x(T \in \cdot) Q(\mathrm{d}x)$.

Intuitively speaking, pufferfish privacy is the requirement that the derivative of $M_{\text{Ext}}$ is bounded, in just the same way that pure DP is the requirement that the derivative of $M$ is bounded (Bailie et al., 2026a; Bailie & Gong, 2024). More exactly, pufferfish privacy constructs a metric $d_{\mathcal{P}}$ on the space $\mathcal{P}$ of probability distributions on $(\mathsf{X}, \mathcal{X})$ in the following way: Given a set of data-priors $\mathcal{Q}$ and a set of competing conjectures $\mathbb{S} \subset \mathcal{X} \times \mathcal{X}$, let $G$ be the graph on $\mathcal{P}$ which has edges between the distributions $Q(X \in \cdot \mid X \in E)$ and $Q(X \in \cdot \mid X \in E')$ for all $Q \in \mathcal{Q}$ and all $(E, E') \in \mathbb{S}$. Define $d_{\mathcal{P}}(P, Q)$ as the length of the shortest path between $P$ and $Q$ on $G$. We say that a mechanism $M$ satisfies $\varepsilon$-pufferfish if $d_{\text{Mult}}[M_{\text{Ext}}(P), M_{\text{Ext}}(Q)] \leq \varepsilon d_{\mathcal{P}}(P, Q)$, where $d_{\text{Mult}}[M_{\text{Ext}}(P), M_{\text{Ext}}(Q)] = \sup_{A \in \mathcal{T}} \left| \log \frac{\mathbb{P}_P(T \in A)}{\mathbb{P}_Q(T \in A)} \right|$ is the multiplicative distance between the distributions of $M_{\text{Ext}}(P)$ and $M_{\text{Ext}}(Q)$.[11]

There are multiple variants of pufferfish privacy (see Bailie et al., 2026a, and references therein), including multiple notions of "Bayesian" DP (Yang et al., 2015; Triastcyn & Faltings, 2020; Aliakbarpour et al., 2025). Some of these variants replace $d_{\text{Mult}}$ with other distances (or, more generally, premetrics—see Bailie et al., 2026a). Others generalise the definition of $d_{\mathcal{P}}$, or consider specific instantiations of $d_{\mathcal{P}}$. For example, profile-based differential privacy (PBDP, Geumlek & Chaudhuri, 2019) considers an arbitrary neighbouring graph $G$ on $\mathcal{P}$ in place of Pufferfish's graph (which has a specific structure based on the competing conjectures $\mathbb{S}$ and the given data-priors $\mathcal{Q}$ under consideration). Hence, PBDP is a straightforward generalisation of pufferfish privacy.

Because of their Bayesian modelling of $x$, pufferfish privacy and its variants have parallels with the Receiver's view of $x$ in persuasive privacy. Yet they are fundamentally different in how their use of Bayesian modelling relates to their conceptualisation of privacy. Pufferfish and its variants are conditions on the extended mechanism $M_{\text{Ext}}$. They view privacy as stability of $M_{\text{Ext}}$: an unstable $M_{\text{Ext}}$ (i.e., a $M_{\text{Ext}}$ with a large derivative) has low privacy, while a stable $M_{\text{Ext}}$ (i.e., a $M_{\text{Ext}}$ with small derivative) has high privacy. Pufferfish privacy uses its data-priors to modify the stability of the data release mechanism. For example, data-priors that have dependence between records will typically decrease stability and data-priors that are highly uncertain will increase stability. In contrast, the data-priors in persuasive privacy are simply ways to measure "information gain" or "privacy loss". This is because the data-priors are only used by persuasive privacy to calculate the prior and posterior privacy scores. Persuasive privacy is a semantics-first approach as its guarantees are directly in terms of Sender's loss due to Receiver's action. In contrast, pufferfish privacy and its variants enforce indistinguishability between neighbouring data-priors—a condition which does not immediately translate to how Sender can be harmed by the data release.

Persuasive and pufferfish privacy are not immediately reconcilable as the persuasive privacy considers the privacy loss for each $x$ (see Assumption 5), while pufferfish privacy and its variants marginalise over $x$. However, modifications of Assumption 5 analogous to QIF (see Appendix D.1) may resolve this difference. This suggests that pufferfish privacy could be understood from the perspective of an adjusted version of persuasive privacy (along the lines of Proposition 6).

As alluded to in the introduction, while the vast majority of DP flavours cannot assess the privacy leakage of deterministic mechanisms, there are some that can. Most of these are pufferfish flavours. For example, noiseless DP (Bhaskar et al., 2011) is a specialisation of pufferfish privacy. Even though deterministic mechanisms $M$ are not stable and as such cannot satisfy regular DP definitions, noiseless DP and other pufferfish variants can handle deterministic mechanisms because a data-prior's uncertainty can make $M_{\text{Ext}}$ stable even when $M$ is not. This is not to say that the output $T$ of $M_{\text{Ext}}$ is not very informative about the true value of the sensitive dataset $x$; but simply that the attacker had sufficient uncertainty a-priori to outweigh the informativeness of $T$. In contrast, persuasive privacy's reasoning about deterministic mechanisms is very different: A deterministic mechanism can satisfy a persuasive privacy definition if, intuitively, Receiver does not learn too much, or is not able to cause much additional harm from observing $T$. This excludes mechanisms with outputs that are informative about $x$, since these would result in a large difference between prior and posterior privacy scores. A deterministic mechanism—or any mechanism, for that matter—can only satisfy persuasive privacy if, simply put, its output is not that informative (with respect to the scoring rule $S$).

---

[11]This formulation of pufferfish privacy is due to Bailie & Gong (2024), which also proves the equivalence between this formulation and the original formulation given in Kifer & Machanavajjhala (2014).

# E. Proofs Deferred from the Main Text

## E.1. Worst-Case Data-Averaged Outcome

*Proof of Proposition 1.* Under Assumption 3, where $\ell = \rho$, Sender's average privacy value is $\mathbb{E}_{X \sim P}[\rho(d_\rho^P, X)]$, satisfying $\mathbb{E}_{X \sim P}[\rho(d_\rho^P, X)] \leq \mathbb{E}_{X \sim P}[\rho(d, X)]$ for all $d \in \mathcal{D}$. Considering some alternative loss function $\ell \neq \rho$, Receiver's optimal decision is

$$d_\ell^P \in \arg\inf_{d \in \mathcal{D}} \mathbb{E}_{X \sim P}[\ell(d, X)] \subset \mathcal{D},$$

and the optimal set is assumed to be non-empty. Therefore, $\mathbb{E}_{X \sim P}[\rho(d_\rho^P, X)] \leq \mathbb{E}_{X \sim P}[\rho(d_\ell^P, X)]$ as $d_\ell^P \in \mathcal{D}$, indicating that Receiver having loss function $\ell = \rho$ is the worst-case data-averaged privacy value for Sender. $\qquad\square$

## E.2. Composition Rule

*Proof of Proposition 3.* We denote Receiver's data-posterior after sequential updating with $T_1$ then $T_2$ by

$$Q_{T_1} = Q(\,\cdot\mid M_1, T_1), \text{ and } Q_{T_1, T_2} = Q(\,\cdot\mid M_1 \otimes M_2, (T_1, T_2)),$$

respectively. Since $\mathcal{Q}_x$ is conjugate to $M_1$ we can state that $Q_{T_1} \in \mathcal{Q}_x$ for $Q \in \mathcal{Q}_x$ almost surely. Then, since $M_2$ is $(\mathcal{S}, \mathcal{Q}_x, \kappa_2, \delta_2)$-PP for all $T_1 \in \mathsf{T}_1$, we have that

$$\mathbb{P}_x\left[\Delta_S(Q_{T_1}, T_2, x) \leq \kappa_2 \mid T_1\right] = \mathbb{P}_x\left[S(Q_{T_1}, x) - S(Q_{T_1, T_2}, x) \leq \kappa_2 \mid T_1\right] \geq 1 - \delta_2, \tag{13}$$

for all $Q \in \mathcal{Q}_x$, $x \in \mathsf{X}$, and $S \in \mathcal{S}$. The requisite probability satisfies

$$\begin{aligned} \mathbb{P}_x\left[\Delta_S(Q, (T_1, T_2), x) \leq \kappa_1 + \kappa_2\right] &= \mathbb{P}_x\left[S(Q, x) - S(Q_{T_1, T_2}, x) \leq \kappa_1 + \kappa_2\right] \\ &= \mathbb{P}_x\left[S(Q, x) \leq S(Q_{T_1, T_2}, x) + \kappa_1 + \kappa_2\right] \\ &\geq \mathbb{P}_x\left[S(Q, x) \leq S(Q_{T_1}, x) + \kappa_1 \leq S(Q_{T_1, T_2}, x) + \kappa_1 + \kappa_2\right] \\ &\geq \mathbb{P}_x\left[S(Q, x) \leq S(Q_{T_1}, x) + \kappa_1\right] + \mathbb{P}_x\left[S(Q_{T_1}, x) \leq S(Q_{T_1, T_2}, x) + \kappa_2\right] - 1, \end{aligned}$$

using a Boole-Fréchet inequality for the last step. Simplifying, using (13) under expectation of $T_1 \sim M_1(x, \cdot)$, and by assumption of privacy for $M_1$, $\mathbb{P}_x\left[\Delta_S(Q, T_1, x) \leq \kappa_1\right] \geq 1 - \delta_1$, we can find that

$$\begin{aligned} \mathbb{P}_x\left[\Delta_S(Q, (T_1, T_2), x) \leq \kappa_1 + \kappa_2\right] &\geq \mathbb{P}_x\left[\Delta_S(Q, T_1, x) \leq \kappa_1\right] + \mathbb{E}_{T_1 \sim M_1(x, \cdot)}\mathbb{P}_x\left[\Delta_S(Q_{T_1}, T_2, x) \leq \kappa_2 \mid T_1\right] - 1 \\ &\geq 1 - (\delta_1 + \delta_2), \end{aligned}$$

for all $Q \in \mathcal{Q}_x$, $x \in \mathsf{X}$, and $S \in \mathcal{S}$. $\qquad\square$

## E.3. Receiver Post-Processing

*Proof of Proposition 4.* Receiver's data-posterior after observing $(T_1, T_2) \sim M \otimes K$ is $Q_{T_1, T_2}$. Yet, since $K$ is independent of the data, no new information about $x$ has been incorporated, hence it is clear that

$$Q_{T_1, T_2} = Q(\,\cdot\mid M \otimes K, (T_1, T_2)) = Q(\,\cdot\mid M, T_1) = Q_{T_1},$$

in this case. Therefore,

$$\begin{aligned} \mathbb{P}_x\left[\Delta_S(Q, (T_1, T_2), x) \leq \kappa\right] &= \mathbb{P}_x\left[S(Q, x) - S(Q_{T_1, T_2}, x) \leq \kappa\right] \\ &= \mathbb{P}_x\left[S(Q, x) - S(Q_{T_1}, x) \leq \kappa\right] \\ &= \mathbb{P}_x\left[\Delta_S(Q, T_1, x) \leq \kappa\right] \\ &\geq 1 - \delta, \end{aligned}$$

for all $Q \in \mathcal{Q}_x$, $x \in \mathsf{X}$, and $S \in \mathcal{S}$ by the privacy assumption for $M$. $\qquad\square$

## E.4. Sender Post-Processing

*Proof of Proposition 5 by counterexample.* Any probabilistic differential privacy (PDP) guarantee is an instance of a persuasive privacy guarantee by Proposition 6. Further, a PDP guarantee does not satisfy sender post-processing (i.e., the post-processing inequality, Kifer & Lin, 2012; Meiser, 2018). $\qquad\square$

### E.5. Probabilistic Differential Privacy

*Proof of Proposition 6.* For all $Q \in \mathcal{H}$ we define $L$ as in Definition 9 (Appendix A) and use the following convention. For $\Delta_L(Q, T, x) = L(Q, x) - L(Q_T, x)$, when $L(Q, x) = L(Q_T, x) = \infty$, we take $\Delta_L(Q, T, x) = \infty - \infty = 0$.

Let $Q \in \mathcal{P}_2$ and $M$ be a mechanism. Consider $\mu$ (depending on $Q$) a dominating measure for both the distribution of $M(x, \cdot)$ and $M(x', \cdot)$ and denote by $m(x, \cdot)$ and $m(x', \cdot)$ their corresponding conditional densities respectively. We have $Q(\{x\} \mid T) = 0$ if $Q(\{x\}) = 0$ and otherwise

$$Q(\{x\} \mid T) = \frac{m(x, T)w}{m(x, T)w + m(x', T)(1 - w)},$$

for some $w \in (0, 1)$ where $x, x'$ are the supporting points of $Q$. Then $M$ satisfies $(L, \mathcal{H}, \varepsilon, \delta)$-PP if and only if

$$\inf_{x \in \mathsf{X}} \inf_{Q \in \mathcal{H}} \mathbb{P}_x \left[ -\log Q(\{x\}) + \log Q(\{x\} \mid T) \leq \varepsilon \right] \geq 1 - \delta. \tag{14}$$

Since if $Q(\{x\}) = 0$, then $-\log Q(\{x\}) + \log Q(\{x\} \mid T) = 0$ by convention, for a given $x \in \mathsf{X}$ we can restrict consideration to $Q(\{x\}) > 0$, which implies that (14) is equivalent to

$$\inf_{x \sim x'} \inf_{w \in (0,1)} \mathbb{P}_x \left[ \log \left( \frac{m(x, T)}{m(x, T)w + m(x', T)(1 - w)} \right) \leq \varepsilon \right] \geq 1 - \delta. \tag{15}$$

As $\log \left( \frac{m(x,T)}{m(x,T)w + m(x',T)(1-w)} \right)$ is convex in $w$, we can express the LHS of (15) as

$$\inf_{x \sim x'} \inf_{w \in (0,1)} \mathbb{P}_x \left[ \log \left( \frac{m(x, T)}{m(x, T)w + m(x', T)(1 - w)} \right) \leq \varepsilon \right] = \inf_{x \sim x'} \mathbb{P}_x \left[ \max \left\{ 0, \log \frac{m(x, T)}{m(x', T)} \right\} \leq \varepsilon \right]$$

$$= \inf_{x \sim x'} \mathbb{P}_x \left[ \log \frac{m(x, T)}{m(x', T)} \leq \varepsilon \right],$$

where the second equality follows from $\varepsilon \geq 0$. Therefore (14) is equivalent to

$$\inf_{x \sim x'} \mathbb{P}_x \left[ m(x, T) \leq \exp\{\varepsilon\} m(x', T) \right] \geq 1 - \delta,$$

or $(\varepsilon, \delta)$-PDP. The converse holds analogously. $\qquad \square$

### E.6. Private Empirical Average

*Proof of Proposition 7.* Let $\mu(P)$ be the mean of a distribution $P$, $\Sigma(P)$ be the covariance, $\mu_i(P)$ and $\sigma_i^2(P)$ be the $i$th marginal mean and variance respectively. For convenience, denote $\mu = \mu(Q)$, $\Sigma = \Sigma(Q)$, $\mu_i = \mu_i(Q)$, and $\sigma_i^2 = \sigma_i^2(Q)$ for the data-prior $Q$. Let $\sigma = [\sigma_1 \cdots \sigma_n]^\top$, the column vector of marginal variances from the data-prior.

For $n \geq 2$, if $Q \in \mathcal{G}_x^r$, the release of $\bar{x}$ by Sender yields a data-posterior $Q_{\bar{x}}$ that is a (degenerate) Gaussian distribution with support on the subspace $\{z \in \mathbb{R}^n : \bar{z} = \bar{x}\}$ with mean and variance

$$\mu(Q_{\bar{x}}) = \mu + \frac{\Sigma u}{u^\top \Sigma u}(\bar{x} - \bar{\mu}), \quad \Sigma(Q_{\bar{x}}) = \Sigma - \frac{\Sigma u u^\top \Sigma}{u^\top \Sigma u},$$

where $u$ is a vector of length $n$ such that $u = \frac{1}{n}[1 \cdots 1]^\top$. If we parametrise the prior variance by $\Sigma_{ij} = \rho_{ij}\sigma_i\sigma_j$ for $\rho_{ii} = 1$ and $|\rho_{ij}| < 1$, then the marginal data-posterior mean and variance can be expressed as

$$\mu_i(Q_{\bar{x}}) = \mu_i + \sigma_i \frac{v_i}{v}(\bar{x} - \bar{\mu}), \quad \sigma_i^2(Q_{\bar{x}}) = \sigma_i^2 \left( 1 - \frac{v_i^2}{v} \right),$$

respectively, where $v_i = \frac{1}{n} \sum_{j=1}^n \rho_{ij}\sigma_j$ and $v = \frac{1}{n} \sum_{i=1}^n \sigma_i v_i$.

Before continuing, we establish some inequalities involving $v_i$ and $v$. If $\Phi$ is the correlation matrix $[\Phi]_{ij} = \rho_{ij}$, then the $v_i^2$ and $v$ terms can be written as

$$v_i^2 = \frac{1}{n^2}(e_i^\top \Phi \sigma)^2, \quad v = \frac{1}{n^2} \sigma^\top \Phi \sigma,$$

where $e_i$ is the standard unit vector in the $i$th direction. First note that $n^2(v - v_i^2) = \gamma_i^\top \Phi \gamma_i \geq 0$ where $\gamma_i = \sigma - (e_i^\top \Phi \sigma) e_i$, and hence

$$v \geq v_i^2. \tag{16}$$

Secondly, let $\lambda_1 \geq 1 \geq \lambda_n \geq 0$ be the largest and smallest eigenvalues of $\Phi$ respectively,[12] then

$$n^2(v - v_i^2) = \gamma_i^\top \Phi \gamma_i \geq \lambda_n \|\gamma_i\|_2^2 = \lambda_n \left[ \sum_{j \neq i} \sigma_j^2 + (\sigma_i - e_i^\top \Phi \sigma)^2 \right] \geq \lambda_n \sum_{j \neq i} \sigma_j^2.$$

Using this when $\lambda_n > 0$ (by assumption of positive definiteness of $\Sigma$), we can state that

$$1 - \frac{v_i^2}{v} = \frac{n^2(v - v_i^2)}{n^2 v} = \frac{\gamma_i^\top \Phi \gamma_i}{\sigma^\top \Phi \sigma} \geq \frac{\lambda_n}{\lambda_1} \frac{\sum_{j \neq i} \sigma_j^2}{\|\sigma\|_2^2} \geq \frac{\lambda_n}{\lambda_1} \left( 1 - \frac{\sigma_i^2}{\|\sigma\|_2^2} \right). \tag{17}$$

Considering this data-posterior with the marginal DSS, the $i$th relative privacy score is

$$\Delta_i(Q, \bar{x}, x) = \frac{(\mu_i - x_i)^2}{\sigma_i^2} - \frac{(\mu_i(Q_{\bar{x}}) - x_i)^2}{\sigma_i^2(Q_{\bar{x}})} - \log\left( 1 - \frac{v_i^2}{v} \right).$$

This can be expressed as

$$
\begin{aligned}
\Delta_i(Q, \bar{x}, x) &= \frac{\left(1 - \frac{v_i^2}{v}\right)(\mu_i - x_i)^2 - \left(\sigma_i \frac{v_i}{v}(\bar{x} - \bar{\mu}) + \mu_i - x_i\right)^2}{\left(1 - \frac{v_i^2}{v}\right)\sigma_i^2} - \log\left(1 - \frac{v_i^2}{v}\right) \\
&= \frac{-\frac{v_i^2}{v}(\mu_i - x_i)^2 - \sigma_i^2 \frac{v_i^2}{v^2}(\bar{x} - \bar{\mu})^2 - \frac{2 v_i \sigma_i}{v}(\mu_i - x_i)(\bar{x} - \bar{\mu})}{\left(1 - \frac{v_i^2}{v}\right)\sigma_i^2} - \log\left(1 - \frac{v_i^2}{v}\right) \\
&= \frac{-v_i^2(\mu_i - x_i)^2 - \sigma_i^2 \frac{v_i^2}{v}(\bar{x} - \bar{\mu})^2 - 2 v_i \sigma_i(\mu_i - x_i)(\bar{x} - \bar{\mu})}{(v - v_i^2)\sigma_i^2} - \log\left(1 - \frac{v_i^2}{v}\right) \\
&= \frac{-[v_i(\mu_i - x_i) + \sigma_i(\bar{x} - \bar{\mu})]^2 + \sigma_i^2(1 - \frac{v_i^2}{v})(\bar{x} - \bar{\mu})^2}{(v - v_i^2)\sigma_i^2} - \log\left(1 - \frac{v_i^2}{v}\right) \\
&\leq \frac{(\bar{x} - \bar{\mu})^2}{v} - \log\left(1 - \frac{v_i^2}{v}\right) \\
&\leq \frac{(\bar{x} - \bar{\mu})^2}{v} - \log \frac{\lambda_n}{\lambda_1}\left(1 - \frac{\sigma_i^2}{\|\sigma\|_2^2}\right),
\end{aligned}
$$

noting $v - v_i^2 \geq 0$ as established by (16) and using (17). Since $v = \overline{\Sigma}$ and $c_\Phi = \lambda_1/\lambda_n$, we find

$$\Delta_i(Q, \bar{x}, x) \leq r_1 + \log r_2,$$

as $\frac{(\bar{x} - \bar{\mu})^2}{\overline{\Sigma}} \leq r_1$ and $c_\Phi \left(1 - \frac{\sigma_i^2}{\|\sigma\|_2^2}\right)^{-1} \leq r_2$ by assumption on the class of data-priors. $\qquad\square$

### E.7. Private Cell Suppression

*Proof of Proposition 8.* After observing the output $T \sim \prod_{i=1}^n M_i^h(x_i, \cdot)$ from the cell-suppression mechanism, Receiver's $i$th marginal data-posterior probability mass function (PMF) $q_i(\cdot \mid T)$, evaluated at the data $x_i$, satisfies

$$q_i(x_i \mid T) = \begin{cases} \frac{q_i(x_i)}{F_i^h} & \text{if } x_i < h, \\ 1 & x_i \geq h, \end{cases}$$

---

[12]$\lambda_1 \geq 1$ by the Schur-Horn Theorem.

where $q_i(\cdot)$ is the $i$th marginal data-prior probability mass function, and the normalising term $F_i^h = Q_i(\{z \in \mathbb{N} : z < h\})$ is simply Receiver's prior probability that the $i$th cell has a value less than $h$. The $i$th relative privacy score is

$$\Delta_i(Q, T, x) = L_i^r(Q, x) - L_i^r(Q_T, x) = \begin{cases} -\log F_i^h & \text{if } x_i < \min\{r, h\}, \\ -\log q_i(x_i) & \text{if } h \leq x_i < r, \\ 0 & \text{if } x_i \geq r, \end{cases}$$

almost surely. Taking $h \geq r$, we can say

$$\mathbb{P}_x[\Delta_i(Q, T, x) \leq \kappa] = \begin{cases} 1[F_i^h \geq e^{-\kappa}] & \text{if } x_i < r, \\ 1 & \text{if } x_i \geq r. \end{cases}$$

Hence, if $\kappa = -\log \alpha$, privacy is retained in all cases with $Q_i \in \mathcal{C}^{h,\alpha}$. □

