# OpenReview forum: "Persuasive Privacy"
_ICML.cc/2026/Conference — ICML 2026 regular_

### Official Review · Reviewer_BzWq · 2026-03-09

**Soundness:** 3
**Presentation:** 3
**Significance:** 3
**Originality:** 4
**Overall Recommendation:** 5
**Confidence:** 4

**Summary:**

This paper analyzes privacy using a framework similar to *Bayesian Persusion*, which models the process of statistical disclosure as a Stackelberg Game between a Sender (the data curator) and a Receiver (the adversary). In particular, the harm caused by the Receiver is modeled by the payoff from the action they choose.

The main contributions of the paper are in formulating this problem/framework and connecting it to more popular formulations of privacy such as differential privacy (DP) through the definition of Persuasive Privacy (Definition 2).

**Compliance With Llm Reviewing Policy:**

Affirmed.

**Final Justification:**

As I wrote, I am happy with the proposed revisions and look forward to reading the final version.

**Key Questions For Authors:**

I found this paper quite interesting because I had coincidentally been reading a little bit about Bayesian Persuasion after learning about it from a prior collaborator of mine around a year ago, although not in the context of privacy. I have a few questions/comments regarding the model and its applicability (which I think is the key weakness in the paper from a "moving forward" perspective).

* In general, it would be nice to have some non-toy examples to help illustrate the kinds of applications where this framework would be appropriate.
* Similarly, a short example of when composition holds or does not hold for PP would help build some intuition.
* One thing that sets DP apart is the lack of stochastic assumptions on the data. Here there is an explicit data prior $Q$. Are there ways to make some of the formulation universal over $Q$ or perhaps to relax the assumption on knowing $Q$ exactly?
* Can we think of Assumption 2 as more or less WLOG? That is, is the most harmful action (worst case) found through Bayesian estimation?
* Definition 1 (on privacy functions) is more like a "harm function" than a "privacy function" -- to my understanding it is the payoff of the game, essentially. I think something more descriptive would help since this is quantifying the cost from privacy loss w.r.t. some external measure, not the privacy loss itself.
* Are there connections between this framework and that of Quantitative Information Flow (QIF)?
* In Assumption 4 shouldn't this be a formal definition of $(\kappa,\delta)$ privacy or something like that?
* The shift from utility to harm-reduction is interesting but also we miss out on the standard tradeoff of privacy and utility. I realize this was explicitly left for future work but some comments on what such results might look like would be helpful.

**Limitations:**

Yes

**Strengths And Weaknesses:**

**Strengths:**

* To the best of my knowledge this is the first work to consider connections between existing privacy definitions and Bayesian persuasion.
* Focusing on modeling the harms from the Receiver's actions is a nice way to operationalize the cost of privacy.

**Weaknesses:**

* The power/usefulness of the proposed framework is not really clear since it's more about definitions than application.
* The string Bayesian modeling might be a bit restrictive (although this might be unavoidable).

---

> ### Author Rebuttal · Authors · 2026-03-31
>
> We thank the reviewer for the careful reading and the positive assessment of the originality and soundness of our work. We would like to address the concern that the framework's power and usefulness are not sufficiently clear.
>
> We agree there were limited worked examples in the original version of the paper (we address this in the camera-ready version, see response Q3, Rev. pSjd, and W1, Rev. ANE1). Our framework directly provides a semantics-first approach: each assumption can be independently scrutinised or relaxed. We recover pure DP, probabilistic DP, and Rényi DP (see response Q1, Rev. BTn2), in addition to providing new agent-based interpretations of the post-processing inequality.
>
> Responses to questions:
>
> 1. We will include a further example in the camera-ready version of the manuscript relevant to official statistics agencies, please see our response to Rev. pSjd (Q3).
>
> 2. Composition holds for DP since a family of two-component discrete priors is conjugate to any stochastic mechanism, for example. As a non-example, suppose we consider the class of priors from the mean release in Section 5. If we release the mean, the Receiver's posterior will be outside the prior class considered (it is degenerate on a subspace defined by knowledge of the true mean, having infinite condition number). Considering the composition of a second mechanism (satisfying the same privacy guarantee in Section 5) with the mean release therefore won’t satisfy the conjugacy condition.
>
> 3. Whilst we do have an assumption on data priors (a family of priors for which one is held by Receiver), we do not have ‘stochastic assumptions on the data’. Rather, just like DP we consider the worst case data set over the domain of data. Hence, we are able to recover DP as part of our framework. It is possible to consider non-parametric families of priors, though we have left this for future work.
>
> 4. This is a very interesting idea. It is possible that we can leverage the minimax–Bayes rule duality since we consider the worst case under the prior (in a given family of priors). However, the proof of this is complicated by the use of the relative privacy score. We leave this for future work.
>
> 5. The harm function terminology would also be suitable. We are inclined to keep privacy because we are “measuring privacy” where privacy is a positive attribute of a release mechanism.
>
> 6. There are similarities between QIF and persuasive privacy. Both assume transparency in guarantees and mechanism code (like DP). They both define privacy in terms of an attacker’s loss (or dually, “gain” in QIF terminology), under the assumptions that the attacker is Bayesian and that they take their Bayes optimal action. Moreover, they both view this privacy loss as relative–i.e., they both compare loss before and after the mechanism’s release.
>
> However, there are substantial differences. In QIF, the posterior loss is an average of the optimal losses under $Q_T$, averaged over the marginal distribution of T – i.e., the posterior loss equals $\mathbb E_T [ \inf_{d \in D} \mathbb E_{X \sim Q_T} [ \ell (d, X) ] ]$. As proponents of QIF readily point out (see [1]), this averaging makes QIF potentially inadequate for assessing privacy. One approach to rectify this would be to average first over the conditional distribution of T given x, and then consider the worst-case over x. The approach we take in the current paper is to look at the tail probabilities of the (unaveraged) relative privacy loss. We view this approach as more robust. It provides stronger guarantees, because we ensure worst-case relative privacy loss is bounded (while allowing for the possibility of a small \delta failure probability).
>
> Both our framework and QIF allow for robustness in two other ways, by requiring uniform bounds over both 1) multiple different data priors, and 2) multiple different privacy loss functions.
>
> 7. We designate Assump. 4 and 5 as assumptions rather than definitions, since they are elements of the privacy setup we consider that can be interpreted, relaxed or changed. Doing so will lead to an alternative family of privacy definition to Persuasive Privacy. Assump. 4 could be viewed mathematically as a general definition, but it is not strictly speaking a privacy definition under Assump. 1 due to dependence on x.
>
> 8. Incorporating statistical utility in the framework is straight-forward. We do also briefly mention this in footnote 6 on p.5. Since a given privacy definition generates a privacy class (for which all mechanisms are ‘equally’ private) one natural approach would be to choose amongst this class (or a parametric subset of this class, for example) the best mechanism according to some utility, such as statistical efficiency. This aligns with utility considerations for DP mechanisms in the literature.
>
> [1] https://www.uni-muenster.de/IFIP-WG22/Web/meeting/Lisbon13/boreale.pdf

---

> > ### Author Rebuttal · Reviewer_BzWq · 2026-03-31
> >
> > I think the additional context will help and my immediate questions are resolved. This hopefully will spark some interesting discussions!

---

> > > ### Author Response · Authors · 2026-04-03
> > >
> > > Thank you once again for your constructive feedback and for improving the paper. We will include this additional context to the camera-ready version. We also hope to see some interesting discussion coming out of the paper -- of course, if you have any extra comments, we would be very grateful to hear them.

---

### Official Review · Reviewer_ANE1 · 2026-03-10

**Soundness:** 3
**Presentation:** 2
**Significance:** 2
**Originality:** 3
**Overall Recommendation:** 4
**Confidence:** 4

**Summary:**

The paper proposes a framework for privacy called persuasive privacy, building on a Bayesian game between the data holder (sender) and an adversary (receiver). The authors show that this notion of privacy exhibits certain interesting properties, such as composition and post-processing at the receiver, under several assumptions. Persuasive privacy is also shown to generalize the pure and approximate notions of differential privacy. Further, it is shown that their notion can be used to assess privacy for deterministic algorithms. The authors show that the empirical average function satisfies persuasive privacy under assumptions on the framework and the distribution of data points.

**Compliance With Llm Reviewing Policy:**

Affirmed.

**Final Justification:**

My concerns were addressed in the rebuttal. I will maintain my score considering the scope and presentation.

**Key Questions For Authors:**

1. How sensitive are the results stated in the paper to the assumption of Bayesian rationality on the receiver and the transparency of the sender? As for assumption 3,  the optimal choice of loss function for a Bayes rational receiver, can the authors provide an intuitive/operational explanation for it to match the privacy function of the sender?

2. How are the choices of the score set and the prior family class constrained? If one can choose them freely, when does PP behave as a privacy guarantee rather than mere model assumptions? Can one make many mechanisms look 'private' under your framework by picking a forgiving score or a prior class?

3. How restrictive is the conjugacy condition used to prove the composition property of PP? Without conjugacy, can you still say something about conjugacy?

4. Page 3, Column 2, Paragraph below Assumption 1 - In the example, it is not clear to me which privacy guarantee can be satisfied by a constant mechanism that completely reveals the data? Wouldn't such a privacy notion be a very bad one?

5. Potential Typos:
Line 74 (column 2): We use of proper --> we use proper
Line 82 (Column 2): I recommend using a different symbol, say S' for expected score, instead of using S, which is already used for the scoring rule.
Line 89 (Column 2): construct a new class --> construct new class
Appendix A, Line 614, Definition 8, log Q({x})---> - logQ({x})
Appendix C.6, Line 847, r1 - log r2 ---> r1 +log r2

**Limitations:**

yes

**Strengths And Weaknesses:**

Strengths:

1. The game-theoretic approach to developing a notion for privacy is quite novel. The fact that it satisfies the common properties of a privacy measure makes it interesting.

2. The persuasive privacy generalizes on the \epsilon-differential privacy and (\epsilon,\delta)-differential privacy. The analysis provides a semantic re-interpretation of DP in terms of bounding the change from prior to posterior distribution under a specific score function and a prior family.

3. The applicability of persuasive privacy to a deterministic mechanism is something that DP lacks. Even though I am a bit skeptical about its practical meaning, it is a worthwhile theoretical result.

Weakness:

1. The framework of persuasive privacy hinges on several modelling assumptions (even on the adversary-receiver), ranging from transparency on the sender, Bayesian rationality on the receiver, etc.

2.  The framework in general seems too flexible in the sense that a user can choose a score set \cal{S} and a prior class \cal{Q}_x, under which several guarantees can be made true or false. A natural question to think about here is, how are these choices constrained?

3. The paper claims that its privacy framework establishes guarantees for deterministic mechanisms. However, the result is proved only for a specific deterministic function (empirical mean) and under restrictive assumptions on the receiver's Gaussian prior. So, it feels like the result is basically saying that the mechanism is private only against a bounded family of not-too-strong adversaries. In my opinion, this restricts the notion of privacy as an algorithm should be either private or non-private, irrespective of how powerful the adversary/receiver is.

4. As the paper argues, PP has easier semantics than DP; however, its parameters could be hard to explain. For eg, a practitioner who only cares about epsilon and delta in DP needs to care about the scoring rule, adversary's prior class, threshold kappa, and delta. This makes the PP framework less portable in practice, in my opinion, due to the large communication complexity. I would like to hear the opinion of the authors in this context.

---

> ### Author Rebuttal · Authors · 2026-03-31
>
> Thank you for the thorough, constructive review.
>
> Weaknesses:
> 1. We view explicitness as a feature rather than a limitation: Our central motivation is that semantic interpretations of DP have been constructed post hoc and are difficult to justify or interrogate, whereas our assumptions have a clear operational meaning that can be scrutinised and adapted.
> 2. We agree this flexibility requires care, though note that DP can equally be misused through vacuous parameter choices.
> 3. The constraint on the prior family in Sec. 5 does not solely limit the strength of the adversary. The first condition limits Receiver’s weakness. The second condition ensures that there is no ‘collapse’ in the posterior (e.g., knowing the mean reveals some coordinates in the data vector). In the case of releasing the mean, still routinely done with no privacy guarantee, our result informs which adversary will ‘break’ privacy. Further, we can improve this in several directions, e.g. by considering a trimmed mean mechanism, allowing for different Receivers.
> 4. We view PP as a meta-definition that will yield (after further work) specific tailored instantiations for use when DP is not flexible enough (e.g., deterministic cases). In particular releasing robust estimators (in the statistical sense) can lead to PP without invoking a randomized step.
>
>
> Questions:
> Q1. Sender is assuming Receiver is Bayes rational to represent Receiver’s optimal decision and how it will affect Sender’s privacy. Other decision-making paradigms are possible (minimax criterion). There may be a link with minimaxity as we do consider the worst case prior Receiver can hold, but this is beyond our scope. Still, we make all assumptions behind PP explicit, so that future work can change or relax them, as signaled in our discussion.
>
> It is difficult to justify not being transparent about the mechanism because privacy guarantees should be given to external parties to verify them. Still, our framework is very flexible: Consider the mechanism $f(x) + z$ where $z\\sim N(0,s)$ where s is hidden (i.e., unknown to the Receiver). The variance s can be included as pseudo-data in x (it is unknown) and specified as part of Receiver's prior. The privacy score used would ignore the variance term (because its privacy need not to be protected).
>
> The formal motivation for Assump. 3 is given in Prop. 1; if the loss function is equal to the privacy function, it corresponds to the data-averaged worst case outcome for Sender’s privacy. Intuitively, if Receiver chooses the privacy function as their loss then they are directly targeting privacy, rather than some other objective. We include Assump. 3 explicitly to indicate that other choices are possible (see also response to Q1, Rev. pSjd).
>
> Q2. There is a lot of flexibility and hence potential for misuse in our framework. Note that DP can also be misused through vacuous parameter choices.
>
> Specific privacy scores and prior classes can lead to poor privacy guarantees. The extreme example is the class of priors containing only a point mass. In this case the prior will never update and privacy will always be preserved. However, it will be clear from the communication of this definition that the privacy definition is contrived and should not be trusted. Instantiations of PP can and should always be tested for appropriateness in the context they are deployed. This is good practice for the use of any privacy definition. Practically, the class of priors should be chosen to be as large as possible whilst remaining realistic for the given scenario. We will add this to our discussion in the camera-ready version of the paper.
>
> Q3. There is a small mistake in the statement of Prop. 3, there is no requirement for the class of priors to be conjugate to $M_2$ (only $M_1$ is required). This has been corrected. As for the conjugacy for $M_1$, enlarging the class of priors will lead to stronger privacy. As such, if the class of priors is not conjugate to $M_1$ it can be enlarged so that it is. Then the composition rule can be used. Alternatively, one could prove a PP guarantee for the class of priors implied by the first posterior update. Applying the conjugacy condition to DP; two-component discrete priors are conjugate to any stochastic mechanism.
>
> Q4. Here we assume the family of mechanisms parametrized by the data cannot depend on the actual data x implicitly. For example, the mechanism defined by $M_x(z,.) = \\delta_x$ depends on x implicitly, and will have no privacy, but our definition would not detect it as we consider data dependence only through the first argument of M. Thus, we explicitly prohibit such implicit dependence of M on the data with Assump. 1. The paragraph you identify attempts to explain this. We will clarify this in the final version.
>
> Q5. Thank you, these typos are fixed. Using S for both a scoring rule and an expected score is standard (e.g., Gneiting & Raftery, 2007) with the second argument determining which is being considered.

---

> > ### Author Rebuttal · Reviewer_ANE1 · 2026-04-03
> >
> > satisfied with answers

---

> > > ### Author Response · Authors · 2026-04-04
> > >
> > > Thank you again for your valuable feedback. The points you raised in your review have improved our work (in particular, by supplying the new, additional examples of censored bins in histograms/cell suppression [see our response to Reviewer pSjd] and of trimmed means).
> > >
> > > We are glad that you are satisfied with our answers to your questions and weaknesses. We are definitely incorporating these answers in the camera-ready version of the manuscript.
> > >
> > > In light of your indication that your concerns have been fully addressed, we would gratefully appreciate it if you might consider whether the paper is now closer to an “accept” in your view. Should any concerns remain, we would greatly appreciate even a brief indication and would be happy to address them.
> > >
> > > Regardless, we thank you once again for your review and constructive criticism.

---

### Official Review · Reviewer_pSjd · 2026-03-12

**Soundness:** 3
**Presentation:** 3
**Significance:** 3
**Originality:** 3
**Overall Recommendation:** 4
**Confidence:** 4

**Summary:**

This paper proposes Persuasive Privacy, a game-theoretic framework that defines privacy through a Stackelberg interaction between a data holder and a Bayesian receiver. The framework is used to derive purpose-specific privacy guarantees, recover pure and probabilistic differential privacy as special cases, establish a composition result and a receiver-side post-processing property, and analyze a deterministic release example. The paper is ambitious, mathematically careful, and conceptually interesting, but its practical validation is very limited.

**Compliance With Llm Reviewing Policy:**

Affirmed.

**Key Questions For Authors:**

1、The framework critically relies on the assumption that the receiver’s loss coincides with the sender’s privacy function. How essential is this assumption to the main results, and what parts of the framework would remain valid if this alignment only held approximately rather than exactly?
2、A central claim of the paper is that it enables fit-for-purpose privacy notions. In a concrete application, how should one choose the privacy function, the scoring rule family, and the admissible prior class in a principled and reproducible way?
3、The deterministic mechanism discussion is presented as an important motivating case, but the paper only analyzes one illustrative example under fairly specific assumptions. Can the authors provide stronger evidence that the framework yields meaningful nontrivial guarantees for a broader class of deterministic releases?

**Limitations:**

1、The paper is almost entirely conceptual and theoretical, and the lack of empirical validation makes it difficult to assess its practical usefulness or interpretability.
2、The framework depends heavily on a strong alignment assumption between the sender’s privacy notion and the receiver’s decision objective, which may not hold in many realistic threat models.
3、While the framework is presented as flexible, the paper does not yet provide a sufficiently concrete methodology for selecting privacy functions, scoring rules, and adversary prior classes in real applications.

**Strengths And Weaknesses:**

Strengths
1、The paper engages with a genuine and important foundational question: whether privacy definitions should be derived from an explicit semantic model of adversarial inference, rather than introduced axiomatically and only interpreted after the fact.
2、The theoretical development is internally coherent, and the paper does a reasonably good job of connecting its game-theoretic setup, Bayesian decision model, scoring-rule construction, and privacy guarantee into a single formal narrative.
3、Even though the paper is not fully convincing in all respects, the central perspective is thought-provoking, and the proposed semantics-first view of privacy could motivate useful follow-up work on application-specific privacy notions and deterministic release settings.
Weaknesses
1、The paper provides almost no empirical validation, so its broader claims about usefulness, interpretability, or practical relevance are not supported beyond formal derivations and a single illustrative deterministic example.
2、The central assumption that the receiver’s loss coincides with the sender’s privacy function is very strong, and the paper does not sufficiently analyze how sensitive the framework is to departures from this alignment.
3、The deterministic mechanism section is too narrow to support the strength of the paper’s broader claims, since it focuses on a single illustrative setting under fairly specific assumptions rather than demonstrating a more general and robust phenomenon.
4、Much of the framework’s claimed flexibility remains at the level of promise rather than evidence, because the paper does not systematically show how different privacy functions or scoring rules would materially change the resulting guarantees in realistic use cases.
5、The paper makes me feel like a draft that has not yet been fully polished. There are several noticeable language and grammatical errors throughout the text, such as “that will effect the privacy of Sender” and “We use of proper scoring rules”. Careful proofreading will help improve the overall presentation and readability of the paper.

---

> ### Author Rebuttal · Authors · 2026-03-31
>
> We thank the reviewer for their positive comments and their interest in the game theoretic perspective we take to privacy.
>
> Weaknesses:
> 1. A further example is provided below (see Q3 below), which we believe is more realistic and is in the final version. This example considers censored bins in histograms, and the setup is more broadly applicable to cell-suppression in contingency tables. Our aim is to lay the foundations of a new approach to privacy, where different setups are considered, including DP. We leave to future (and ongoing) work thorough, empirical validation of specific instantiations of Persuasive Privacy (PP) motivated by real-world contexts similar to the histogram example below. Such work is beyond the scope of the present paper.
> 2. Our justification from Prop. 1 supports Assump. 3 (see Q1 below). The identified weakness is also shared by DP (as a special case of PP) but, to the best of our knowledge, this view is not supported by the literature.
> 3. See Q3 below.
> 4. We agree that this is a limitation of the paper and we have added to the discussion to better acknowledge this. We expect material validation of PP to be on a case-by-case basis. A substantial contribution we do make is delineating between two different types of post-processing (see response to Rev. BTn2).
> 5. We have corrected these typos and other minor errors after carefully reviewing the paper.
>
> Questions:
>
> Q1. Sender’s privacy function and Receiver's loss function coinciding (Assump. 3) is supported by Prop. 1, which interprets this assumption as the data-averaged worst case for privacy. Alternatives to Assump. 3 are possible, but these will result in a different class of privacy definitions. For example, if one can assume the worst-case over a class of loss functions for Receiver. As the reviewer suggests, it is also possible to define such a class of Receiver loss functions as deviations from the privacy function. The obvious downside is the loss of the scoring rule duality which is convenient analytically.
>
> Q2. Our framework is deliberately generic (as in decision theory) so that privacy can be fit for purpose. Since another example proves beneficial, we added it in the final version (see Q3), which we hope addresses the reviewers concern about illustrating concrete applications.
>
> Q3. The deterministic mechanism example (mean release) is conceptually interesting because it yields a degenerate posterior (constrained to a subspace of the original support) for Receiver. Typically, any degeneracy implies the prior-to-posterior privacy bounds are indeterminate or infinite, in most existing privacy definitions. However, use of the marginal DSS (Def. 8) shows it is possible to circumvent this issue. This insight is an important technical contribution for future examples, currently beyond the scope of this paper. A further concrete example of the usefulness of PP is as follows. Consider releasing a histogram with fixed bin sizes (similarly a contingency table with one row). We assess the deterministic mechanism that releases the histogram with a bin’s value suppressed if it contains a count of k or less. The data is $x \in \\{0,1,...\\}^n$. Receiver will hold a prior over the counts in each bin, here assumed to be independent. If bin i has more than k items, say $x_i$, Receiver’s posterior based on the histogram is a point mass on $x_i$.  When $x_i \\leq k$, Receiver’s posterior for this bin is the original prior truncated to {0,...,k}. The bound k parametrizes the mechanism, higher k will result in more privacy (from censoring).
>
> The most natural privacy scores to use are proper categorical or binary scores. Statistical agencies are often concerned with adversarial knowledge that less than m subjects satisfy some condition, possibly revealed by a histogram or table. Here the privacy concern is to hide bins with low counts, as a possible information leak about individuals. One choice of score for the $i$th marginal (of the distribution P) is $S’_i(P,x)=1-[1-S_i(P, x)]1(x_i \\in C)$, where C = {1,2,...,m} and $S_i(P, x)=[P(x_i\\in C)-1(x\\in C)]^2+[P(x_i\\notin C)-1(x_i\\notin C)]^2$. The $S_i(P,x)$ term is a grouped (binary quadratic) Brier score for each bin, whilst $S’_i(P,x)$ is altered to always be 1 (privacy retained) when $x_i \\notin C$. The quantity m parametrizes the privacy score through the set C. That is, the agency is not concerned with protecting privacy for bins with counts outside of C. If considering the relative privacy score, Sender will be controlling the change in prior-to-posterior probability of knowledge that the $i$th bin has a count $x_i \\in C$. Then a natural question for the agency is what value of k is required so that the posterior update after observing the censored histogram doesn’t reveal too much information about the censored bins. Noting that larger k will result in the posterior approaching the prior for the censored bins.
>
> We include this as a further example in the final version.

---

> > ### Author Rebuttal · Reviewer_pSjd · 2026-04-07
> >
> > The authors did their best to resolve the issues, and I will maintain my score due to their core contribution.

---

> > > ### Author Response · Authors · 2026-04-07
> > >
> > > Thank you for your constructive criticism and comments during the review process. The paper has improved thanks to your input.

---

### Official Review · Reviewer_BTn2 · 2026-03-13

**Soundness:** 2
**Presentation:** 4
**Significance:** 3
**Originality:** 3
**Overall Recommendation:** 4
**Confidence:** 4

**Summary:**

This paper re-states privacy in the form of a two-player Stackelberg game between a sender and receiver; by playing with the proper loss function between the sender the paper shows that different kinds of privacy guarantees such as differential privacy may be obtained. They also show that persuasive privacy mechanisms have certain properties related to composition and post-processing invariance.

**Compliance With Llm Reviewing Policy:**

Affirmed.

**Final Justification:**

I am supportive of accepting this paper (given that the authors add the necessary discussion).

**Key Questions For Authors:**

see above

**Limitations:**

yes

**Strengths And Weaknesses:**

+ Paper is quite well-written; there are quite a few nice examples, and definitions are explained well and clearly

+ The concept while abstract makes sense, and a couple of natural privacy definitions do fall out of it


- The main weakness of this paper is the lack of connections to existing privacy literature. There have been many many variants of differential privacy that have been proposed already; a successful alternative notion or definition would be expected to make connections with them.

While it is true that this is a short conference paper, it would be nice to have more of those connections here -- some examples are:

-- Pufferfish [ Macchanavajhhala et al, 2014]
-- Capacity-bounded differential privacy [ Chaudhuri, Imola and Machhanavajhhala, 2019 ]
-- Profile-based Privacy [ Geumlek et al, 2019 ]
-- Coupled world privacy [ Bassily et al, 2014 ]

---

> ### Author Rebuttal · Authors · 2026-03-31
>
> We thank the reviewer for the positive assessment and the constructive feedback. Though not our main goal, we agree that connections to the broader privacy literature are valuable, and attempt to address these below.
>
> **(1)** There are many variants of differential privacy (DP) already proposed in the literature but we are not attempting here to propose another variant of DP. Our goal is to build a new framework, distinct from DP, explicitly motivated and constructed from the ground up using the tools and perspectives of game theory. It therefore seems somewhat contradictory to seek more connections with alternate DP definitions. Furthermore, such connections appear to be nontrivial (because our foundations and ideas fundamentally differ from DP’s, as expanded on below).
>
> Nevertheless, the reviewer’s question of how Persuasive Privacy (PP) fits broadly within the existing literature on statistical data privacy is a valid one. Some answers are to be found in the related work section (Sec. 1.1), including connections to variants of DP such as Pufferfish (mentioned by the reviewer). Due to space consideration, we were unable to expand further on these answers. However, we show that pure DP and probabilistic DP (PDP), canonical definitions of data privacy fit within PP. While other definitions have been developed to address shortcomings in DP, the focus therein required a significant focus on DP. An important understanding our framework gains, but not acknowledged in the reviews, is that PP – and therefore PDP – does satisfy a type of post-processing invariance, contrary to current understanding in the literature. Our framework enables delineation between two types of post-processing (Receiver vs Sender post-processing, RPP and SPP respectively) and elucidates why PDP only satisfies the former. Further, we expose that RPP is the version of post-processing that is important from a privacy point, whereas SPP is a convenient (but often lossy) tool for proving privacy guarantees. We also demonstrate how to avoid the problem with SPP altogether for PP.
>
> Making connections to other privacy definitions is possible, but can require different assumptions to those presented in the paper. Our previous reluctance to include further connections is due to emphasising simplicity in the presentation and lack of space. We can recover Rényi DP (in fact any DP version where the comparison between the distributions of the mechanisms under x and x’ is an f-divergence) by altering Assump. 4. To recover Rényi DP, consider the settings of the PDP example (Prop 6), but rather than controlling the tail probability (as specified by Assump. 4), bound the worst-case expected relative privacy score. Following similar steps to the proof of Prop 6, we will recover Rényi DP with alpha = 1. For general Rényi DP (and f-divergences as in capacity bounded DP) we take the negative exponential of the relative privacy score and apply a convex function, before taking the expectation. The proof relies on the joint convexity property of f-divergences which allows us to determine that the worst-case under the alternative hypothesis prior class is when the weight goes to 0.  We will add this (along with accompanying proof) to the camera-ready version, with a discussion of the merits of considering tail probabilities or expected values when evaluating stochastic privacy scores. We will also mention that connecting to capacity-bounded DP may be possible (as a generalisation of Rényi DP) but is beyond the scope of this paper.
>
> Regarding Pufferfish privacy, whilst we think it would be possible to find a game-theoretic semantics-first derivation, we believe that it is unlikely to be a special case of PP without substantial changes to the assumptions we have. Our reasoning for this is as follows. Pufferfish privacy involves assessing the difference between input-averaged mechanisms where the averaging component differs (see Bailie & Gong, 2024). Though we are able to change Assump. 4 to handle expectations (i.e., averaging) of the ratio of distributions (e.g., to recover Renyi DP), it is not clear how to extend this to the case where the averaging is done on the two mechanisms separately. As Pufferfish privacy is a special case of profile-based privacy, the same reasoning leads us to suspect profile-based privacy is similarly characterized.
>
> We have also drawn substantial comparisons between PP and the Quantitative Information Flow literature on data privacy. See our response to Rev. BzWq. We include these discussions in the camera-ready version of the paper.
>
> **(2)** The reviewer report does not reference any issues with soundness, yet has been scored 2. Is this due to the aforementioned issue? The reviewer also writes “yes” as their response to “Limitations”, without any elaboration. Are we correct in understanding that this is also (solely) referencing the question of connections to other work?
>
> Bailie & Gong, 2024 https://doi.org/10.1016/j.ijar.2024.109242

---

> > ### Author Rebuttal · Reviewer_BTn2 · 2026-04-03
> >
> > As promised please discuss these points in the paper.

---

> > > ### Author Response · Authors · 2026-04-04
> > >
> > > Thank you again for your constructive criticism regarding building additional connections to existing literature. We are glad that the connections we now make have addressed your concerns. We are definitely including them in the paper.
> > >
> > > Given that you have marked your concerns as fully resolved, we wanted to ask—respectfully—whether the paper might now be closer to an "accept" in your assessment. If there are still any remaining concerns that prevent such an update, we would of course be very grateful for a brief indication and would be happy to work towards further improving our paper in that direction.
> > >
> > > In any case, we are grateful for your thoughtful reading and helpful feedback, which improved the paper.

---

### Decision · Program_Chairs · 2026-04-30

**Decision:**

Accept (regular)

**Comment:**

The authors construct a "semantics-first" framework where privacy definitions are derived from explicit agent motivations rather than axiomatic assumptions. Key highlights include:

- Generalization of DP: It demonstrates that pure and probabilistic differential privacy (DP) are special cases of this framework.
- Deterministic Mechanisms: Unlike standard DP, which requires randomization, this framework can establish privacy guarantees for deterministic algorithms (e.g., releasing an empirical mean under specific prior assumptions).
- Post-Processing: It introduces a distinction between Receiver-side and Sender-side post-processing, providing a new interpretation of why certain privacy notions are invariant to these operations.

The reviewers are in alignment about the originality and soundness of the work.

The main downsides of the paper are its limited empirical evaluation and some assumptions being too strong, specifically, the reliance on the adversary (Receiver) being perfectly Bayesian, and the strong alignment required between the Receiver's loss function and the Sender's privacy function.